# Noisy neuronal populations effectively encode sound localization in the dorsal inferior colliculus of awake mice

Juan Carlos Boffi[1,2]*, Brice Bathellier[3], Hiroki Asari[2], Robert Prevedel[1,2,4,5,6]*

[1]Cell Biology and Biophysics Unit, European Molecular Biology Laboratory, Heidelberg, Germany; [2]Epigenetics and Neurobiology Unit, European Molecular Biology Laboratory, Monterotondo, Italy; [3]Université Paris Cité, Institut Pasteur, AP-HP, Inserm, Fondation Pour l'Audition, Institut de l'Audition, IHU reConnect, Paris, France; [4]Developmental Biology Unit, European Molecular Biology Laboratory, Heidelberg, Germany; [5]Molecular Medicine Partnership Unit, European Molecular Biology Laboratory, Heidelberg, Germany; [6]Interdisciplinary Center for Neurosciences, Heidelberg University, Heidelberg, Germany

## eLife Assessment

The paper reports the **important** discovery that the mouse dorsal inferior colliculus, an auditory midbrain area, encodes sound location. The evidence supporting the claims is **solid**, being supported by both optical and electrophysiological recordings. The observations described should be of interest to auditory researchers studying the neural mechanisms of sound localization and the role of noise correlations in population coding.

*For correspondence:
juan.boffi@embl.de (JCarlosB);
robert.prevedel@embl.de (RP)

**Abstract** Sound location coding has been extensively studied at the central nucleus of the mammalian inferior colliculus (CNIC), supporting a population code. However, this population code has not been extensively characterized on the single-trial level with simultaneous recordings or at other anatomical regions like the dorsal cortex of inferior colliculus (DCIC), which is relevant for learning-induced experience dependent plasticity. To address these knowledge gaps, here we made in two complementary ways large-scale recordings of DCIC populations from awake mice in response to sounds delivered from 13 different frontal horizontal locations (azimuths): volumetric two-photon calcium imaging with ~700 cells simultaneously recorded at a relatively low temporal resolution, and high-density single-unit extracellular recordings with ~20 cells simultaneously recorded at a high temporal resolution. Independent of the method, the recorded DCIC population responses revealed substantial trial-to-trial variation (neuronal noise) which was significantly correlated across pairs of neurons (noise correlations) in the passively listening condition. Nevertheless, decoding analysis supported that these noisy response patterns encode sound location on the single-trial basis, reaching errors that match the discrimination ability of mice. The detected noise correlations contributed to minimize the error of the DCIC population code of sound azimuth. Altogether these findings point out that DCIC can encode sound location in a similar format to what has been proposed for CNIC, opening exciting questions about how noise correlations could shape this code in the context of cortico-collicular input and experience-dependent plasticity.

## Introduction

Locating a sound source facilitates essential behaviors such as foraging, mating and predator avoidance, thanks to the omni-directionality and long reach of the auditory system (*King et al., 2001*). In vertebrates, sound localization relies on both binaural cues such as interaural level and time differences (ILD, ITD; *Knudsen and Konishi, 1979*) and monaural cues including spectral notches (SN; *Kulkarni and Colburn, 1998*). These sound localization cues are processed independently at brainstem nuclei in the ascending auditory pathway and integrated altogether for the first time at the inferior colliculus (IC; *Adams, 1979*; *Brunso-Bechtold et al., 1981*; *Gourévitch and Portfors, 2018*; *Grothe et al., 2010*). This makes the IC a crucial early relay of the ascending auditory pathway to study how a primary neural representation of auditory space is formed (*Gourévitch and Portfors, 2018*; *Grothe et al., 2010*). Furthermore, the IC is also targeted by cortico-fugal interactions involved in higher order functions concerning sound localization information like experience-dependent plasticity (*Bajo et al., 2019*; *Bajo et al., 2010*; *Bajo and King, 2012*), supporting that IC plays a complex part in the sound localization processing network, contributing from primary representation to shaping behavior.

Previous studies involving extracellular recordings from the mammalian IC revealed that average responses from IC neurons were tuned to multiple sound location cues, evidencing the integration of such cues at IC (*Chase and Young, 2005*). However, the amount of sound location information carried by individual neurons throughout the auditory pathway is limited and quite variable, suggesting that the sound location information is distributed into a population code for auditory space (*Clarey et al., 1995*; *Day and Delgutte, 2016*; *Groh et al., 2003*; *Panniello et al., 2018*). Fundamental work by *Day and Delgutte, 2013* supports that sound location in the horizontal plane (azimuth) is represented at the mammalian IC by the activity patterns of populations of neurons that display average response tuning to azimuth, while ruling out other possibilities such as a topological code, an opposing two channel model or a population vector code (*Georgopoulos et al., 1986*; *Georgopoulos et al., 1982*). Nevertheless, this knowledge stems from prospective IC population activity patterns that were approximated by pooling non-simultaneous extracellular recordings and relied on extensive resampling of the pooled recordings to increase the number of 'virtual' trials analyzed. Therefore, it remains an open question whether the actual activity patterns occurring in the IC of awake animals during single-trials can encode sound location given emerging properties such as correlated neuronal noise (*Averbeck et al., 2006*; *Kohn et al., 2016*; *Sadeghi et al., 2019*). Trial-to-trial response variability (neuronal noise) and neuronal noise correlation can have a profound impact on the neural representations of sensory information, but this can only be determined through simultaneous large-scale recordings (*Averbeck et al., 2006*; *Kohn et al., 2016*). Finally, traditional microelectrode recordings were mostly performed at a relatively deep anatomical subdivision of IC (central nucleus, CNIC), usually not targeting superficial regions (dorsal cortex, DCIC; external cortex, ECIC; *Chase and Young, 2008*; *Chase and Young, 2005*; *Day and Delgutte, 2013*; *Guo et al., 2016*; *Lesica et al., 2010*; *Schnupp and King, 1997*). To overcome these limitations and explore the importance of trial-to-trial response variability, noise correlation and dorsal IC populations on sound location coding, we monitored DCIC population response patterns on the single-trial basis and interrogated the population code for sound location at the DCIC from passively listening awake mice.

Recent technical developments, including multichannel silicon probes (*Jun et al., 2017*) and fast volumetric $Ca^{2+}$ imaging (*Prevedel et al., 2016*; *Stringer et al., 2019*; *Weisenburger et al., 2019*) enable routine simultaneous recordings from large numbers of neurons in vivo. Beyond their high throughput, volumetric methods are also a promising approach to precisely interrogate the topological distribution of neurons sensitive to sound azimuth, which has been reported to be random at CNIC (*Day and Delgutte, 2013*), but is understudied at DCIC. In this work, we implemented scanned temporal focusing two-photon microscopy (sTeFo 2 P; *Prevedel et al., 2016*), an advanced volumetric $Ca^{2+}$ imaging modality, to simultaneously record the activity of unprecedentedly large DCIC populations. Here, we refer to DCIC as the dorsomedial IC region covering CNIC (*Zhou and Shore, 2006*). Our approach produced direct evidence supporting that the response patterns from mammalian DCIC populations effectively encode sound location on the single-trial basis in spite of their variability across trials. We also detected the occurrence of substantial noise correlations which can contribute to reducing the error of this population code. Furthermore, we complemented our imaging results with electrophysiological recordings with neuropixels probes (*Jun et al., 2017*), reaching conclusions that were generally aligned across imaging and electrophysiological experiments. Altogether, our

findings point to a functional role of DCIC in sound location coding following a similar population coding mechanism to what has been proposed for CNIC (*Day and Delgutte, 2013*). While CNIC is the main ascending relay carrying sound location information from brainstem auditory nuclei to the cortex through the auditory thalamus (*Adams, 1979*; *Brunso-Bechtold et al., 1981*; *Gourévitch and Portfors, 2018*), DCIC is a higher order relay receiving profuse descending corticofugal inputs influencing auditory information processing including sound location (*Bajo et al., 2019*; *Bajo et al., 2010*; *Bajo and King, 2012*; *Lesicko et al., 2022*; *Winer et al., 2002*). This knowledge sets forth exciting possibilities about the mechanisms by which cortico-collicular interactions involving DCIC and perhaps noise correlations affect relevant processes relying on sound location information like experience-dependent plasticity (*Bajo et al., 2019*; *Bajo et al., 2010*; *Bajo and King, 2012*).

## Results

### Simultaneous recordings of DCIC population activity

Since the abundance and distribution of sound localization sensitive neurons at DCIC is not fully characterized, in a first instance we implemented sTeFo-2P for volumetric $Ca^{2+}$ imaging to simultaneously record the activity from samples of DCIC neurons as large as technically possible. Moreover, TeFo-based methods have been shown to be more resilient to highly scattering (optically opaque) tissues such as the IC (*Dana and Shoham, 2012*). Adeno-associated viral vector (AAV) transduced IC neurons expressing jRGECO1a (*Figure 1A*) were imaged through a cranial window in awake head-fixed mice, passively listening to 500 ms-long broad-band noise stimuli (20–40 kHz band-passed white noise) delivered by a motorized speaker every 5 s located at one of 13 different frontal azimuth angles in a random order covering the frontal hemifield in 15° steps (*Figure 1B*). We performed volumetric $Ca^{2+}$ imaging (4 Hz volume rate) simultaneously sampling the activity of up to 2535 (643±427, median ±median absolute deviation, n=12 mice) regions-of-interest (ROIs) that had a shape and size consistent with an IC neuronal soma (12–30 µm diameter, *Figure 1—figure supplement 1A*; *Schofield and Beebe, 2019*), within a 470 x 470 × 500 µm volume from the right IC unilaterally, covering most of the dorso-ventral span of the anterior DCIC (*Figure 1A–C*) and perhaps reaching the upper surface of CNIC below DCIC at more posterior locations (*Paxinos and Franklin, 2001*). Since sTeFo 2 P trades off spatial resolution for acquisition speed to produce volume rates compatible with the kinetics of $Ca^{2+}$ sensors (voxel size: 3.7 x 3.7 × 15 µm; *Prevedel et al., 2016*), we could not resolve fine spatial features of neurons such as dendrites or synaptic structures like spines in our images to unequivocally identify the segmented ROIs as individual neurons (*Figure 1—figure supplement 1A*). For this reason, we refer to these ROIs as 'Units', in an analogous fashion to extracellular electrode recordings studies. To compensate for the relatively slow decay time of the $Ca^{2+}$ indicator signal and non-linearities in the summation of $Ca^{2+}$ events, we evaluated the spike probabilities associated to the neuronal $Ca^{2+}$ signals (*Deneux et al., 2016*; *Figure 1D*, *Figure 1—figure supplement 1B*). The spike probability estimation does not involve predicting precise spike timings (*Deneux et al., 2016*; *Huang et al., 2021*; *Pachitariu et al., 2018*; *Rupprecht et al., 2021*; *Vanwalleghem et al., 2020*). Hence, we do not make any observation or interpretation of spike probabilities in relation to the fine temporal structure of neuronal spiking. Instead, we treated the spike probabilities as a measure proportional to the spike count evaluated within the time interval set by our volumetric imaging rate (4 Hz). Taking into account that the main carrier of sound location information is IC neuron firing rate (*Chase and Young, 2008*), we defined the evoked DCIC population responses as the total (summed) spike probability of each imaged unit during the sound stimulation epoch (500ms). In this way, we systematically collected the simultaneous response patterns of DCIC populations to every azimuth trial.

The DCIC populations simultaneously imaged from passively listening awake mice displayed spontaneous activity that was not time-locked to the stimulation (on-going activity) and sound-evoked response patterns that varied markedly across trials, as did the overall, average response of the complete imaged population (*Figure 1E*). By monitoring face movements videographically, we observed that the overall average population activity shows significant correlation to facial movements, specifically of the snout region (*Figure 1—figure supplement 2A—C*). On the other hand, the correlation between the recorded activity from the individual units imaged and face movements was generally low with narrow distributions centered at 0 (*Figure 1—figure supplement 2D*), pointing out that correlation to face movement is more evident at the population level.

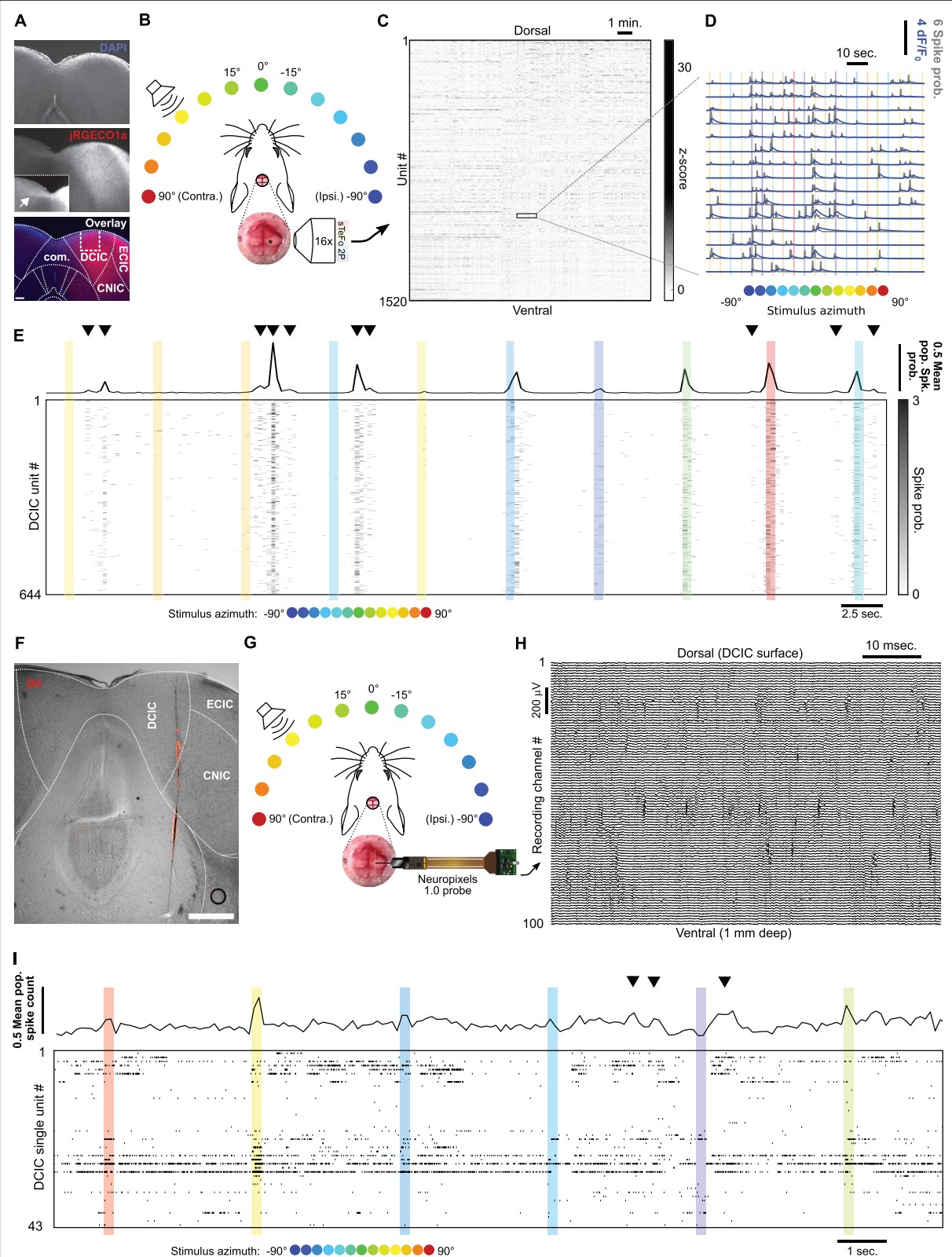

**Figure 1.** Simultaneous recording of DCIC population responses to sound azimuth through sTeFo-2P Ca²⁺ imaging and neuropixels probes.
(**A**) Representative histological section showing AAV transduced jRGECO1a expression across IC. Middle panel inset: Contrast enhanced commissure region of the slice to visualize commissural projections from jRGECO1a expressing IC neurons. Bottom panel: Dotted lines delimit anatomical IC regions according to **Paxinos and Franklin, 2001**; dashed lines delimit approximate area targeted for imaging. Scale bar: 200 µm. DCIC: dorsal

*Figure 1 continued on next page*

*Figure 1 continued*

cortex from inferior colliculus. CNIC: Central nucleus from inferior colliculus. ECIC: External cortex from inferior colliculus. Com.: Commissure from inferior colliculus. (**B**) Schematic representation of the experimental design, incorporating sTeFo 2 P for $Ca^{2+}$ imaging. (**C**) Neuropil-corrected and denoised jRGECO1a signals extracted from a representative full dataset. Extracted signals are arranged from dorsal (top) to ventral (bottom) ROI position across the DCIC volume imaged. (**D**) Representative neuropil corrected and denoised jRGECO1a traces (blue) with their corresponding spike probability traces (gray) and stimulation epochs (color-coded based on stimulus azimuth angle according to (**B**)) super-imposed. (**E**) Representative simultaneous recording of a DCIC population from an awake, passively listening mouse, displaying spontaneous, on-going activity (not synchronized to stimulation, arrowheads) and variable sound-evoked response patterns (during sound stimuli). Top trace is the population average response. Sound stimulation epochs are color-coded based on azimuth. (**F**) Representative histological section showing DiI labeled neuropixels electrode tract across IC. Dotted lines delimit anatomical IC regions according to *Paxinos and Franklin, 2001*. Scale bar: 500 µm. (**G**) Same as (**B**) but representing integration with electrophysiological recording of DCIC population activity with a neuropixels probe. (**H**) Representative high pass filtered (>300 Hz) voltage traces simultaneously recorded from 100 channels spanning across DCIC in a neuropixels probe shank during an experiment displaying clear unit waveforms captured across neighbouring channels. (**I**) Same as (**E**) but showing a representative raster plot of the spike sorted DCIC single-unit activity simultaneously recorded during an experiment.

The online version of this article includes the following figure supplement(s) for figure 1:

**Figure supplement 1.** Examples of extracted signals for imaging and electrophysiology, together with sound frequency tuning at DCIC.

**Figure supplement 2.** Relationship between DCIC on-going activity and face movements.

To further corroborate our volumetric imaging data, we recorded DCIC population activity electrophysiologically through single-unit recordings with neuropixels probes in passively listening mice stimulated with 200 ms-long broad-band noise stimuli (15–50 kHz band-passed white noise) delivered by a motorized speaker every 3 s from 13 different frontal azimuth angles in a random order covering the frontal hemifield in 15° steps (*Figure 1F–H*). The higher time resolution from the electrophysiological approach (30 kHz acquisition rate) allowed us to deliver more frequent trials with shorter stimuli to effectively collect more trials following the same head fixation paradigm as the one used for imaging (see Materials and methods). The electrophysiology setup also produced lower background noise sound pressure level (SPL, 35.96 dB R.M.S.) in comparison to the sTeFo scope (44.83 dB R.M.S.), which enabled the use of a broader band stimulus favoring sound location cues at appropriate levels above background noise (at least 10 dB R.M.S., see Materials and methods). We simultaneously recorded up to 43 (21±9, median ±median absolute deviation, n=4 mice) spike sorted and manually curated DCIC single-units (*Figure 1—figure supplement 1C—E*). We defined sound evoked responses from electrophysiologically recorded single-units as the spike count observed during sound stimulus presentation (*Chase and Young, 2008*; *Day and Delgutte, 2016*; *Day and Delgutte, 2013*). We determined which recording channels from the probe were located within DCIC both histologically with respect to the reference atlas (*Paxinos and Franklin, 2001*; *Figure 1F*) and functionally by analyzing the arrangement of sound frequency sensitive units along the shank of the probe. We detected sound frequency sensitive single-units as units that displayed significant response dependency to sound frequency (pure tone stimuli) through $\chi^2$ tests. The detected frequency dependent single-units displayed Gaussian tuning with clear peak best frequencies (*Figure 1—figure supplement 1F, G*). The observed relationship between frequency-dependent DCIC unit best frequency and recording depth was not tonotopically arranged like in CNIC, which is consistent with DCIC location (*Barnstedt et al., 2015*; *Wong and Borst, 2019*; *Figure 1—figure supplement 1H*). The simultaneous, electrophysiologically recorded DCIC population activity from awake passively listening mice also displayed spontaneous on-going activity, which reflected in the overall average response of the complete recorded population, and sound-evoked response patterns that varied markedly across trials (*Figure 1I*). In this section, we used the word 'noise' to refer to the sound stimuli used, the recording setup background sound levels or recording noise in the acquired signals. To avoid confusion, from now on the word 'noise' will be used in the context of neuronal noise, which is the trial-to-trial variation in neuronal responses unrelated to stimuli, unless otherwise noted.

## Decoding sound azimuth from single-trial DCIC population responses

The observed variability in our imaging and electrophysiological data raised the following question: To what extent do the single-trial responses of the DCIC units carry information about stimulus azimuth? Since the number of DCIC neurons needed to effectively encode sound azimuth is not known, we first evaluated how accurately stimulus azimuth can be predicted from the high-throughput single-trial

DCIC population response patterns volumetrically imaged, without taking resampling strategies or pooling recorded responses across animals or recording sessions as in previous electrophysiological and imaging studies (*Day and Delgutte, 2013*; *Jazayeri and Movshon, 2006*; *Panniello et al., 2018*). We performed cross-validated, multi-class classification of the single-trial population responses (decoding, *Figure 2A*) using a naive Bayes classifier to evaluate the prediction errors as the absolute difference between the stimulus azimuth and the predicted azimuth (*Figure 2A*). We chose this classification algorithm over others due to its generally good performance with limited available data. We visualized the cross-validated prediction error distribution in cumulative plots where the observed prediction errors were compared to the distribution of errors for random azimuth sampling (*Figure 2B*). When decoding all simultaneously recorded units, the observed classifier output was not significantly better (shifted towards smaller prediction errors) than the chance level distribution (*Figure 2B*). The classifier also failed to decode complete DCIC population responses recorded with neuropixels probes (*Figure 3A*). Other classifiers performed similarly (*Figure 2—figure supplement 1A*). Given the high dimensionality of our datasets (number of simultaneously recorded units) and the relatively low number of azimuth presentations collected (10–20 repetitions of each), we reasoned that the classification failure could be due to overfitting rather than to a lack of azimuth information in the dataset. Thus, we tested if reducing the dimension of the dataset helps avoid overfitting. Specifically, we used the same decoder after dimensionality reduction through principal component analysis (PCA) with both PCA and classifier fit being cross-validated (*Figure 2C*). For both our imaging and electrophysiological datasets, we observed that decoding based on subsets of first principal components (PCs) produced absolute cross-validated single-trial prediction error distributions significantly better than the chance level distribution (*Figure 2D and E*, *Figure 3B–D*); and that using larger numbers of PCs for model fitting and decoding decreased the performance (*Figure 2D and E*, *Figure 3C and D*). Altogether, these results support that DCIC population responses indeed carry stimulus azimuth information which can be effectively decoded on a single-trial basis; and that the failure of decoding from the full population was merely due to overfitting.

## Sound azimuth information is carried redundantly by specific DCIC units

The decoded PCs consist of linear combinations of the responses from the units in the imaged or recorded DCIC population. Then the following questions arise: To what extent is the sound azimuth information distributed across the DCIC populations? Is it fully distributed across most units or specific to some of them, such as azimuth tuned neurons?

Since the abundance or distribution of azimuth-sensitive units across DCIC is not fully known, we firstly asked if any fraction of the volumetrically imaged DCIC units carried more azimuth information in their responses than other simultaneously imaged units in the volume. We calculated the signal-to-noise ratio for the responses of each imaged unit ('neuronal' S/N, nS/N), defined as the ratio between the mean (signal) and the standard deviation (noise) of the responses to the azimuth trials evoking maximal responses (best azimuth). Imaged DCIC units generally showed a relatively low nS/N during sound stimulations (<1, *Figure 4A*). Nevertheless, this nS/N was significantly larger than the one registered in the absence of sound stimulation (on-going activity from the periods between sound stimuli, *Figures 1D and 4A*), indicating that sound responses across the sampled units were often noisy. However, the broad distribution of nS/N also suggested that a sparse population of neurons responded more robustly. Next, we evaluated azimuth tuning to delimit subpopulations with higher azimuth selectivity. Using non-parametric one-way ANOVA (Kruskal-Wallis test), we identified the imaged units whose median responses changed significantly based on stimulus azimuth, which we defined as azimuth tuned units (*Figure 4B*). The units with significant azimuth tuning represented only 8 ± 3% (median ±median absolute deviation, n=12 mice) of the DCIC units simultaneously-imaged in one session (mouse; *Figure 4C*). This is not significantly different from the false positive detection rate of our ANOVA tests ($\alpha$=0.05, *Figure 4C*). Given that a large number of trials showed no response at all in many units, it is possible that the large trial-to-trial variability in the imaged DCIC responses made it difficult to detect response specificity based on differences in median activity. We thus employed more sensitive $\chi^2$ tests to determine the statistical dependence between the recorded DCIC responses and presented stimulus azimuth. We then ranked the units in our simultaneously imaged samples based on the p-value of the $\chi^2$ tests (*Figure 4D*) to delimit subsamples of DCIC units showing the

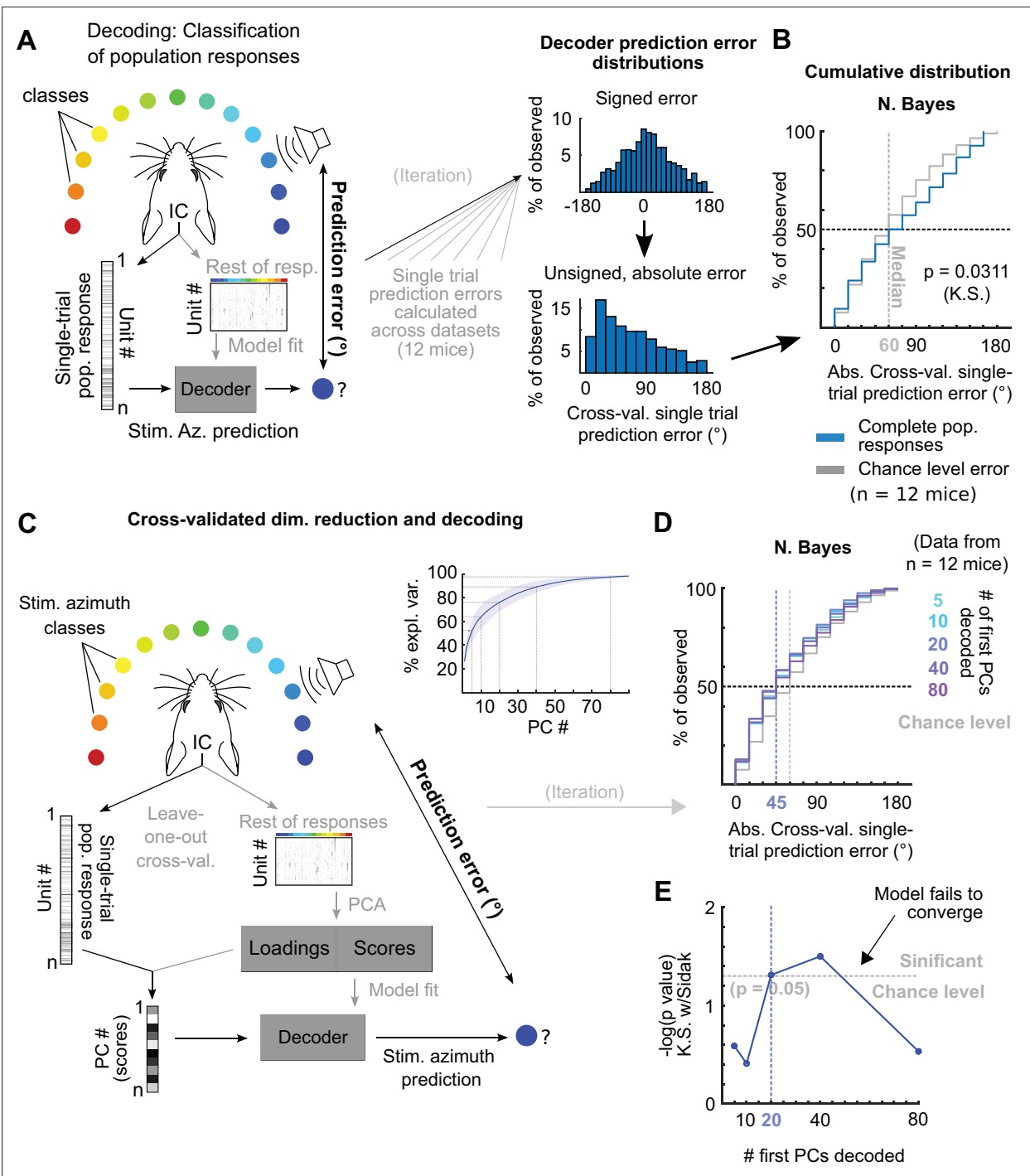

**Figure 2.** Decoding imaged single-trial DCIC population responses to sound azimuth. (**A**) Schematic representation of the decoding strategy using multi-class classification on the recorded simultaneous population responses. (**B**) Cumulative distribution plots of the absolute cross-validated single-trial prediction errors obtained using naive Bayes classification (blue) and chance level distribution associated with our stimulation paradigm obtained by considering all possible prediction errors for the 13 azimuths tested (gray). K.S.: Kolmogorov-Smirnov test, n.s.: (p>0.05). (**C**) Schematic representation of the decoding strategy using the first PCs of the recorded population responses. Inset: % of explained variance obtained using PCA for dimensionality reduction on the complete population responses. Median (blue line) and median absolute deviation (shaded blue area) are plotted for (n=12 mice/imaging sessions). (**D**) Same as (**B**) but for decoding different numbers of first PCs from the recorded complete population responses. (**E**) Significance of classification performance with respect to chance level for different numbers of first PCs, determined via Kolmogorov-Smirnov tests with Sidak correction for multiple comparisons. Arrowhead indicates model loss of performance associated with fitting more parameters for a larger feature space (# PCs) with the same dataset size (# trials collected).

*Figure 2 continued on next page*

*Figure 2 continued*

The online version of this article includes the following figure supplement(s) for figure 2:

**Figure supplement 1.** Alternative decoding models tested.

strongest azimuth-dependent responses. The units with significant response dependency to stimulus azimuth (p<0.05) represented 32 ± 6% (median ±median absolute deviation, n=12 mice) of the units simultaneously imaged in one session (mouse, *Figure 4E*). Even the top ranked units typically did not respond in many trials; nevertheless, their evoked responses showed a tendency to be more selective towards contralateral or central stimulus azimuths (*Figure 4D* left inset). The nS/N of top ranked units was also significantly larger than the one registered in the absence of sound stimulation, but these units did not display a major improvement in nS/N with respect to the complete population of imaged

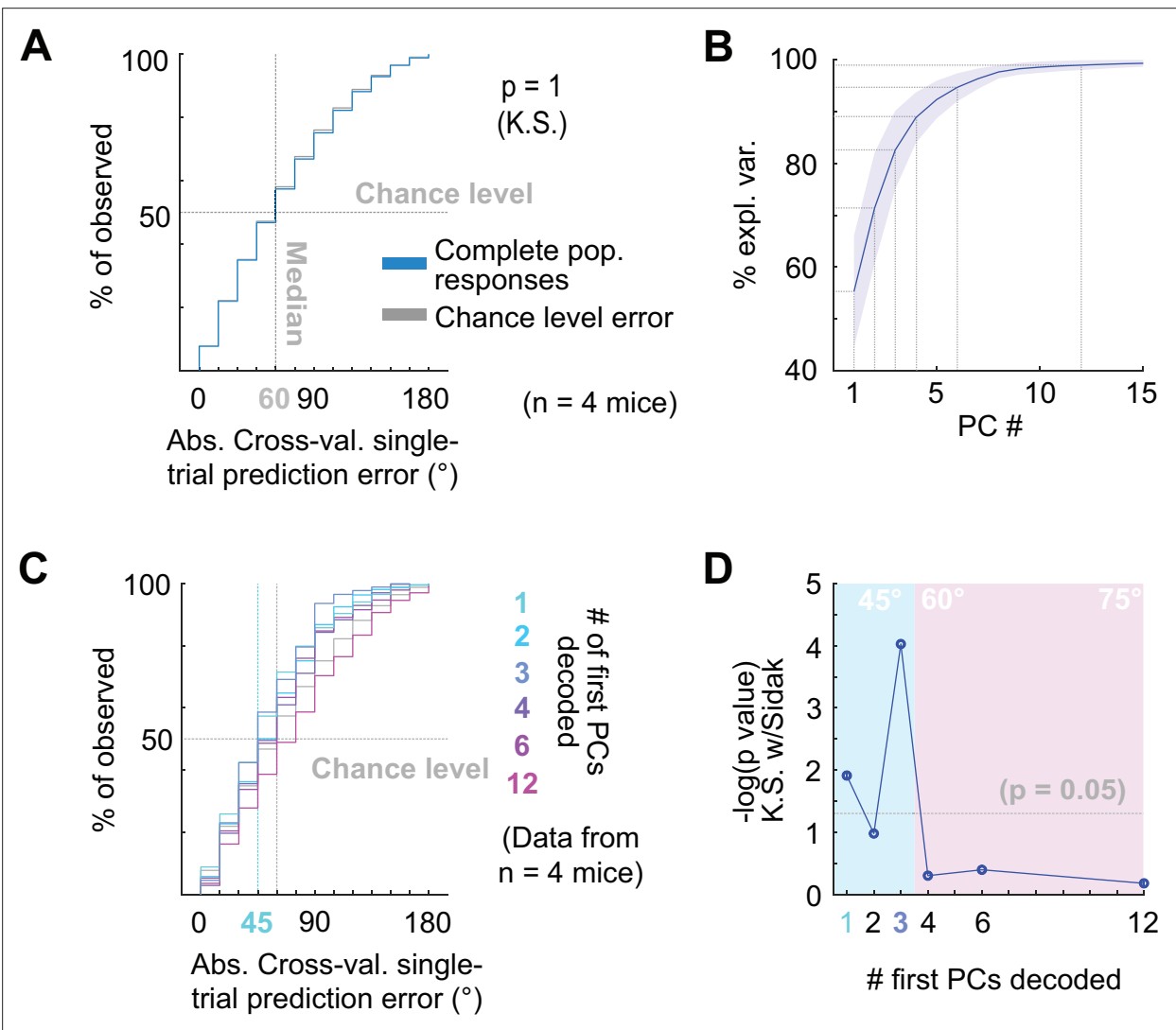

**Figure 3.** Decoding neuropixels recorded single-trial DCIC population responses to sound azimuth. (**A**) Cumulative distribution plots of the absolute cross-validated single-trial prediction errors obtained decoding the complete simultaneously recorded population responses with neuropixels probes across mice (blue) and chance level distribution associated with our stimulation paradigm (gray). K.S.: Kolmogorov-Smirnov test, n.s.: (p>0.05). (**B**) % of observed variance explained across PC number for the complete population responses recorded with neuropixels. Median (blue line) and median absolute deviation (shaded blue area) are plotted for n=4 mice. (**C**) Same as (**A**) but for decoding different numbers of first PCs. (**D**) Significance of decoding performance shown in (**C**) with respect to chance level for different numbers of first PCs, determined via Kolmogorov-Smirnov tests with Sidak correction for multiple comparisons. Shaded areas show the corresponding median decoding errors to the points within the area.

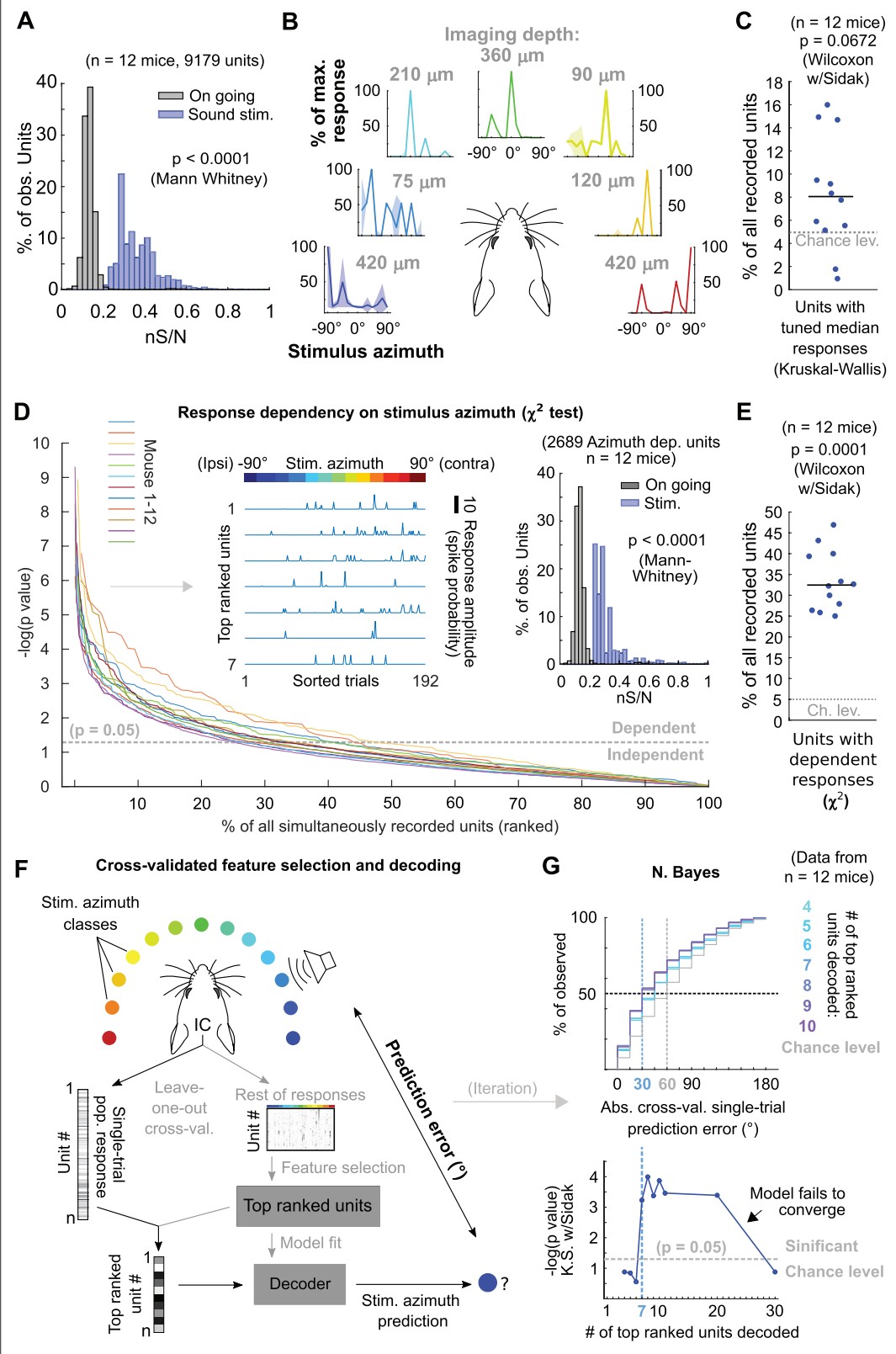

**Figure 4.** Sound azimuth information is carried by specific units from the imaged DCIC populations. (**A**) Histogram of the nS/N ratios from the recorded units across mice during sound stimulation or during the inter trial periods without sound stimulation (on going). (**B**) Representative stimulus azimuth tuning curves from units with significant median response tuning detected using non-parametric one-way ANOVA (Kruskal-Wallis test). Median and

*Figure 4 continued on next page*

*Figure 4 continued*

absolute median deviation are plotted. The imaging depth from the corresponding units is displayed in gray. Azimuth selectivity is color-coded based on *Figure 1B*. (**C**) Percentage of the simultaneously recorded units across mice that showed significant median response tuning, compared to false positive detection rate (α=0.05, chance level). (**D**) Response dependency to stimulus azimuth, determined via $\chi^2$ tests (see Materials and methods), for simultaneously recorded units ranked in descending order of significance. Left inset: Representative responses from the top ranked 7 units with significant response dependency to stimulus azimuth. Response amplitudes are displayed with a continuous trace for visualization purposes, the displayed response order was sorted as a function of stimulus azimuth and does not represent the experimental stimulus delivery order (random). Right inset: Same as (**A**) but for the subset of units displaying response dependency to stimulus azimuth. (**E**) Percentage of the simultaneously recorded units across mice that showed significant response dependency to stimulus azimuth, compared to false positive detection rate (α=0.05, chance level). (**F**) Schematic representation of the decoding strategy using the top ranked units from the recorded population responses. (**G**) Top: Cumulative distribution plot of the absolute cross-validated single-trial prediction errors obtained with a Bayes classifier (N. Bayes, naive approximation for computation efficiency). The number of top ranked units considered for decoding their simultaneously recorded single-trial population response patterns is color coded from cyan (4 top ranked units) to purple (10 top ranked units) and the chance level distribution associated to our stimulation paradigm, obtained by considering all possible prediction errors for the 13 azimuths tested, is displayed in gray. Bottom: Significance of classification performance with respect to chance level for 4–30 decoded top ranked units, determined via Kolmogorov-Smirnov tests with Sidak correction for multiple comparisons. Arrowhead indicates model loss of performance associated with fitting more parameters for a larger feature space (# units) with the same dataset size (# trials collected).

The online version of this article includes the following figure supplement(s) for figure 4:

**Figure supplement 1.** Top ranked units are scattered throughout the imaged DCIC volumes.

(**A**) Histogram plots of the distribution of top ranked unit position in the imaged volume across each anatomical axis (20 µm bins) obtained from all imaged mice, either for the complete sample of units (purple) or the subsample of top ranked units (~32% of all the units imaged per mice, blue). K.S.: Kolmogorov-Smirnov test. (**B**) Scatter plots of the centroid position throughout the anatomical axes in the imaged volume from the detected top ranked units across mice.

---

units, indicating that despite their response dependency to stimulus azimuth their responses are also noisy (*Figure 4A and D* right inset).

To test if the single-trial response patterns from simultaneously imaged units with top response dependency to stimulus azimuth carry enough information to generate better azimuth predictions than the ones observed with the complete samples (*Figure 2*), we evaluated the prediction errors obtained by decoding them. In doing so, we cross-validated both unit ranking based on response dependency (feature selection) and decoder fit (*Figure 4F*). From this point on, we will refer to groups of units showing maximum response dependency to stimulus azimuth as 'top ranked units'. To evaluate the minimum number of top ranked units necessary to generate stimulus azimuth predictions significantly better than the chance level, we decoded different sized subsamples of these units. We found that the single-trial response patterns of at least the 7 top ranked units produced stimulus azimuth prediction errors that were significantly smaller than chance level (p=6 x 10⁻⁴, Kolmogorov-Smirnov with Sidak) using Bayes classification (naive approximation, for computation efficiency; *Figure 4G*). Increasing the number of top ranked units decoded did not bring major improvements in decoder performance (plateau), up to a point where performance dropped, due to classifier overfitting likely caused by the relatively small number of collected trials per class (*Figure 4G*, bottom). If we consider the median of the prediction error distribution as an overall measure of decoding performance, the single-trial response patterns from subsamples of at least the 7 top ranked units produced median decoding errors that coincidentally matched the reported azimuth discrimination ability of mice (*Figure 4G*, minimum audible angle = 31°; *Lauer et al., 2011*).

To further characterize the identified top ranked units, we studied how they are distributed across the imaged DCIC volume, which was positioned roughly at the center from the surface of the mouse dorsal IC (*Figure 1A and B*). By comparing the positions of top ranked units to those of the complete samples of simultaneously imaged units across mice, we observed that the top ranked units (~32% from the complete DCIC populations imaged) are scattered across the imaged volumes following the same distributions across the anatomical axes of the complete imaged populations (*Figure 4—figure*

*supplement 1*). This observation suggests that the subpopulations of top ranked DCIC units associated with the population code of sound azimuth scatter across the DCIC without following a specific spatial pattern. We observed a tendency for a small shift towards positive correlations in the correlation coefficient distribution between snout movement and the imaged activity of individual top ranked units, however this distribution was narrow and centered close to 0 (*Figure 1—figure supplement 2E*).

The observed broad distribution across DCIC and the relatively small number of top ranked units necessary to decode stimulus azimuth supports the characterization of this subpopulation with neuropixels data. Top ranked units detected in our neuropixels datasets also displayed response selectivity towards contralateral and central azimuths and did not respond in many trials (*Figure 5A and B*). The nS/N of the DCIC single-units recorded with neuropixels was again significantly larger than the one registered without sound stimulation, with a distribution tailing above 1. Top ranked units displayed a further shift towards higher nS/N with respect to the complete population of neuropixels recorded DCIC single-units. Still the medians of these nS/N distributions was below 1, supporting that DCIC single-unit responses are noisy (*Figure 5C*). Top ranked units detected with neuropixels recordings represented 40 ± 2% (median ±median absolute deviation, n=4 mice) of the units simultaneously recorded in an experiment (mouse, *Figure 5D*).

Interestingly, the nS/N distributions of neuropixels recorded DCIC single-units in response to pure tone stimuli were similar to those observed for broadband stimulus azimuth (*Figure 1—figure supplement 1I*, *Figure 5C*), suggesting that the observed nS/N levels in our imaging and electrophysiological recordings might be a property of the DCIC network (and not due to the recording method's sensitivity) which is noisy independently of the stimulus presented, at least in the passively listening condition. The percentage of sound frequency dependent units was 42 ± 15% (median ±median absolute deviation, n=4 mice, *Figure 5D*). Out of the DCIC single-units from our neuropixels recordings, 19 ± 6% (median ±median absolute deviation, n=4 mice, *Figure 5D*) displayed significant response dependency to both broadband stimulus azimuth and pure tone stimulus frequency, pointing out that not all DCIC azimuth-dependent units are sensitive to sound frequency and vice versa. To explore a possible relationship between the best frequency and azimuth sensitivity of these units in the context of the duplex theory (*Rayleigh, 1907*), we evaluated the correlation between their best frequency and the significance level of their response dependency to azimuth (-log(p value) of the $\chi^2$ test). This correlation was low (*R*=0.03, 17 single-units from 4 mice), but we could observe that the clear majority of these units had best frequencies above ~10 kHz, where the main sound location cues used by mice are carried (*Lauer et al., 2011*; *Figure 5E*).

Decoding analysis (*Figure 4F*) of the population response patterns from azimuth dependent top ranked units simultaneously recorded with neuropixels probes showed that the 4 top ranked units are the smallest subsample necessary to produce a significant decoding performance that coincidentally matches the discrimination ability of mice (31° (*Lauer et al., 2011*; *Figure 5F and G*)). Altogether, the close resemblance of the results obtained through volumetric imaging and electrophysiologically with neuropixels support that, even though noisy DCIC single-units can encode sound location individually with low performance (*Figure 5F and G*, first top ranked unit), a population code consisting of the simultaneous response patterns from small subsets of top ranked units occurs at DCIC, achieving more effective encoding errors.

## Noise correlations and their contribution to the population code of sound azimuth at DCIC

We next investigated the occurrence of correlated variation in the trial-to-trial responses to each stimulus azimuth from pairs of simultaneously imaged or electrophysiologically recorded top ranked units as the occurrence of such neuronal noise correlations, which have been previously reported to occur at mammalian IC populations (*Sadeghi et al., 2019*), can influence the accuracy of the DCIC population code for sound azimuth (*Averbeck et al., 2006*; *Kohn et al., 2016*; *Figure 6A*). We evaluated the Kendall Tau as an unbiased, non-parametric pairwise correlation coefficient for the neuronal noise from all possible pairs of simultaneously recorded DCIC units across trial repetitions for each stimulus azimuth. Using cross-validated hierarchical clustering of the simultaneously imaged or neuropixels recorded top ranked units based on noise correlation, we observed no clear groups (clusters, subpopulations) of highly noise correlated units across the trial repetitions from each stimulus

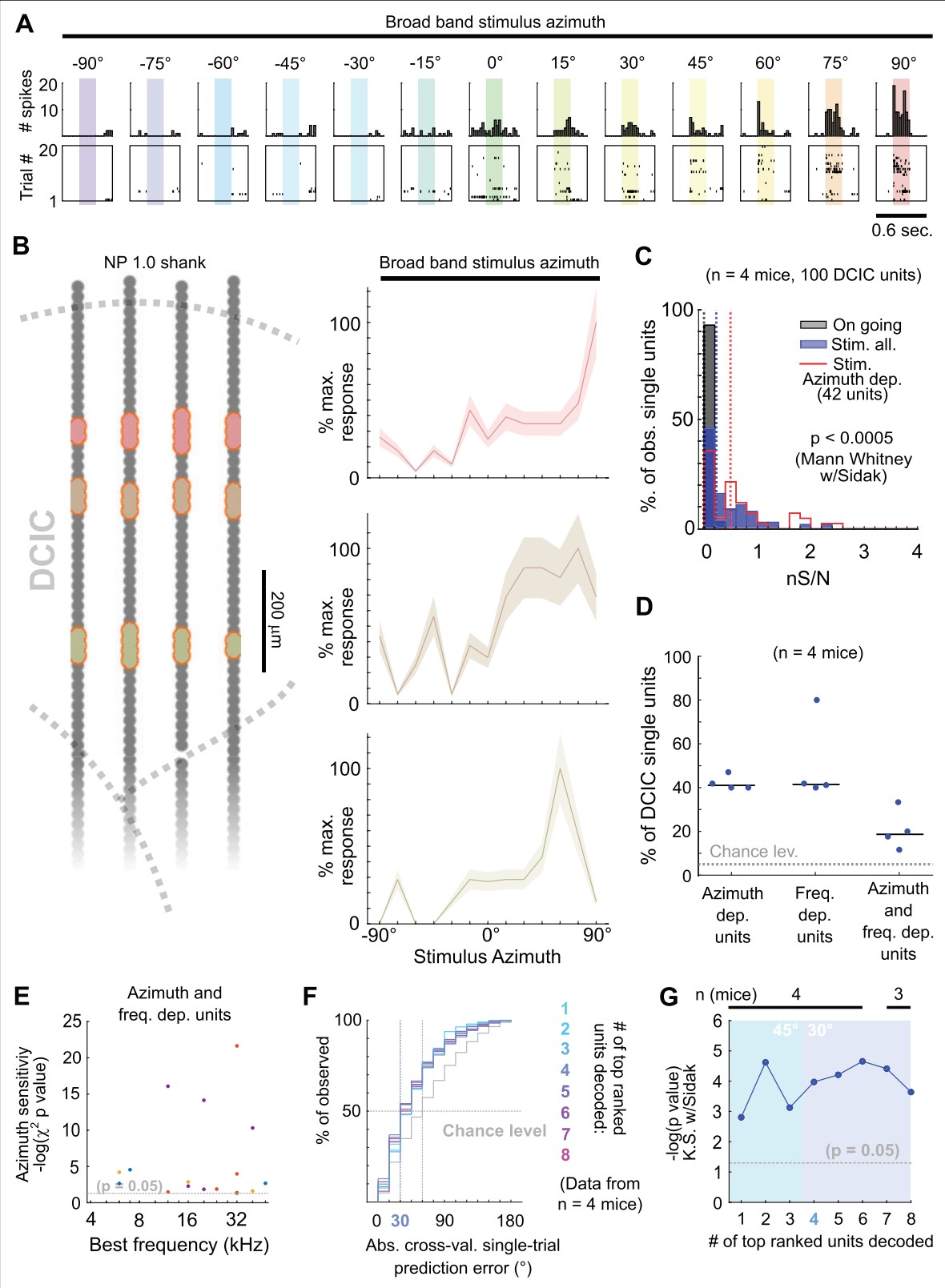

**Figure 5.** Neuropixels recordings support observations drawn from sTeFo 2 P Ca²⁺ imaging experiments. (**A**) Representative responses to stimulus azimuth of a top ranked unit recorded with neuropixels. Top panels show the peri-stimulus time histograms and the bottom panels show the corresponding spike raster plots across trials. (**B**) Left: Schematic representation of a neuropixels probe shank highlighting in different colors the position from channels where representative top ranked single-units were detected across DCIC (approximated histologically, same units as displayed in

Figure 5 continued

**Figure 1—figure supplement 1C—E**). Right Representative azimuth tuning curves from three DCIC top ranked single-units recorded with neuropixels, plot colors correspond to position in the shank schematic. Mean and standard deviation are plotted. (**C**) Neuronal signal to noise level (nS/N) histograms from neuropixels recorded DCIC single-units in the absence of sound stimuli (on going activity, gray) and in response to the best azimuth trials, for all collected single-units (blue) and the top ranked single-unit subset (red). (**D**) Percentages of sound azimuth dependent units (top ranked units), sound frequency dependent units and both azimuth and frequency dependent units across mice. Median value across mice is represented by a horizontal line. (**E**) Relationship between azimuth sensitivity and best frequency of DCIC sound frequency and azimuth dependent single-units across mice. Data from n=4 mice, point color corresponds to the same mouse. X axis scale is logarithmic. (**F**) Cumulative distribution plots of the absolute cross-validated single-trial prediction errors obtained by decoding the responses from different numbers of top ranked units simultaneously recorded with neuropixels probes across mice and chance level distribution associated with our stimulation paradigm (gray). (**G**) Significance of decoding performance shown in (**F**) with respect to chance level for different numbers of top ranked units decoded, determined via Kolmogorov-Smirnov tests with Sidak correction for multiple comparisons. Shaded areas show the corresponding median decoding errors to the points within the area. Sample sizes (number of mice) is informed at the top of the graph for each point.

azimuth (**Figure 6B and C**). However, the pairwise noise correlation coefficients observed across datasets (mice) showed a distribution that was significantly shifted towards positive values with respect to chance level, calculated from the same datasets subjected to randomization of each unit's responses across trial repetitions for each stimulus azimuth (decorrelated, **Figure 6A–C**). Nevertheless, this shift was much smaller for ipsilateral and central azimuths in our neuropixels data (**Figure 6B and C**). Altogether these observations suggest that pairs of the DCIC top ranked units associated with the population code for sound azimuth display positive noise correlations with a likelihood that is higher than chance.

To characterize how the observed positive noise correlations could affect the representation of stimulus azimuth by DCIC top ranked unit population responses, we compared the decoding performance obtained by classifying the single-trial response patterns from top ranked units in the modeled decorrelated datasets versus the acquired data (with noise correlations). With the intention to characterize this with a conservative approach that would be less likely to find a contribution of noise correlations as it assumes response independence, we relied on the naive Bayes classifier for decoding throughout the study. Using this classifier, we observed that the modeled decorrelated datasets produced stimulus azimuth prediction error distributions that were significantly shifted towards higher decoding errors (**Figure 6B and C**) and, in our imaging datasets, were not significantly different from chance level (**Figure 6B**). Altogether, these results suggest that the detected noise correlations in our simultaneously acquired datasets can help reduce the error of the IC population code for sound azimuth. We observed a similar, but not significant tendency with another classifier that does not assume response independence (KNN classifier), although overall producing larger decoding errors than the Bayes classifier (**Figure 2—figure supplement 1B**).

## Discussion

By performing fast volumetric $Ca^{2+}$ imaging through sTeFo-2P microscopy (**Prevedel et al., 2016**) and single-unit recordings with neuropixel probes (**Jun et al., 2017**), here we tackled the technical challenge to simultaneously record the activity of a large number of DCIC units in response to sound azimuth. We show that sTeFo-2P effectively achieved high throughput sampling across large imaging depths in a highly light-scattering tissue such as the mouse IC (**Figure 1**), showcasing the capability of sTeFo-2P for interrogating neuronal population activity over large volumes. Despite the advantages of volumetric sTeFo-2P, it also has limitations. In particular, large-scale volumetric imaging has a low temporal resolution, reaching sampling rates of 4 volumes per second, and requires the use of an indirect method of monitoring neuronal activity via $Ca^{2+}$ sensors. The low temporal resolution effectively produces a low-pass filtering of neuronal activity, misrepresenting peak responses. Furthermore, indirectly inferring neuronal spiking responses from $Ca^{2+}$ sensor signals can cause further information loss. Nevertheless, the main observations drawn from sTeFo 2 P imaged population response patterns could be validated via single-unit recordings with new generation multichannel silicon probes (neuropixels), which have excellent temporal resolution but markedly lower throughput. This supports that the sTeFo imaging datasets still carried sufficient information about stimulus azimuth (**Figures 2 and 4**), showcasing how sTeFo 2 P opens new possibilities for future studies requiring high-throughput recordings of simultaneous population activity, especially considering the advantages brought by

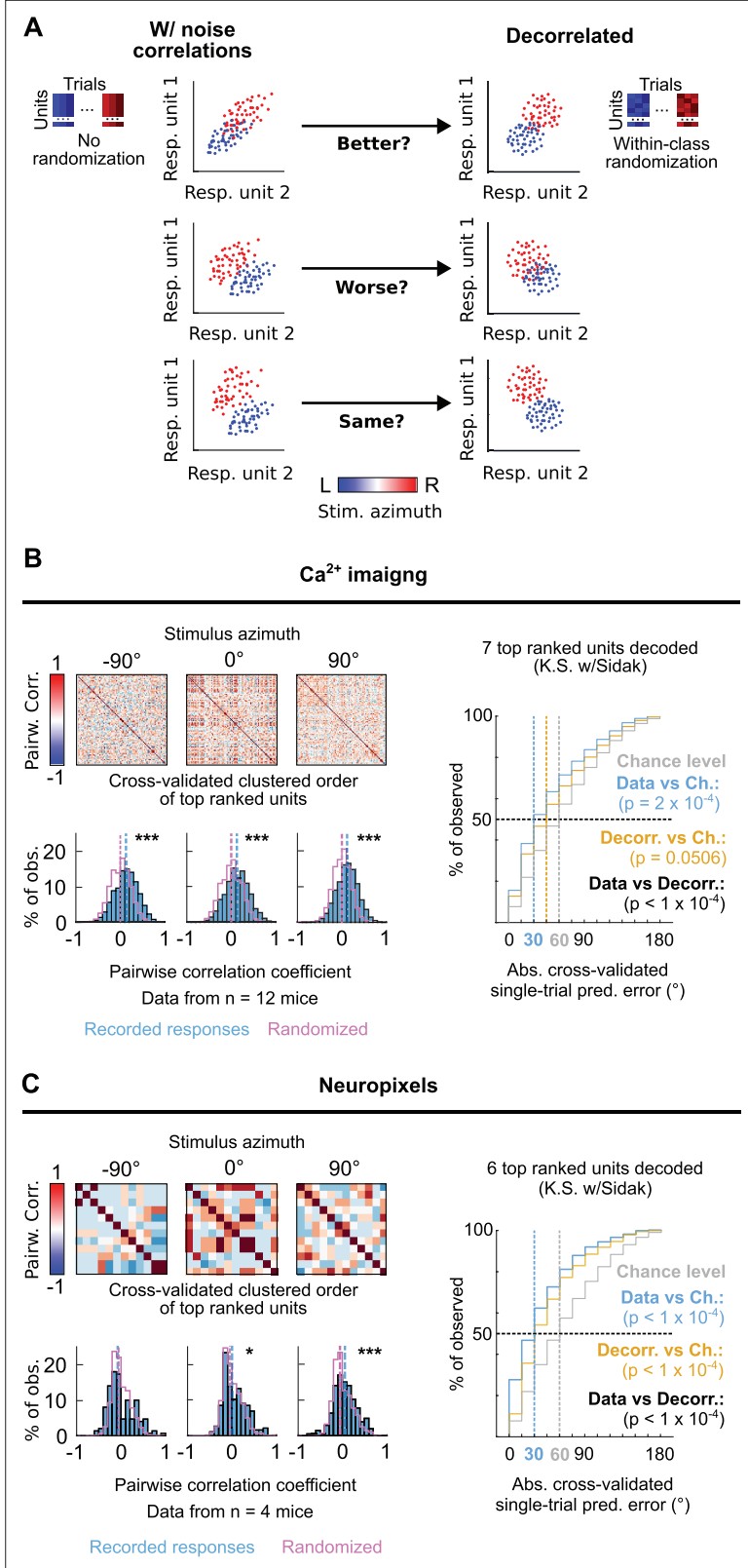

**Figure 6.** Noise correlations in DCIC population activity contribute to encode sound azimuth. (**A**) Simplified schematic representation of the possible effects from (positive) noise correlations on the response separability of a theoretical population consisting of 2 units, and within class randomization strategy to model decorrelated datasets lacking noise correlations. (**B, C**) Left top: Representative correlation matrices of pairwise correlations

*Figure 6 continued on next page*

*Figure 6 continued*

between the responses of top ranked units detected in simultaneous recordings during sound stimuli for representative azimuths. The simultaneously imaged units are sorted in the correlation matrices based on cross validated hierarchical clustering (see Materials and methods). Left bottom: Distribution histograms for the pairwise correlation coefficients (Kendall tau) from pairs of simultaneously recorded top ranked units across mice (blue) compared to the chance level distribution obtained through randomization of the temporal structure of each unit's activity to break correlations (purple). Vertical dashed lines show the medians of these distributions. *: p<0.05, ***: p<0.0001, Kolmogorov-Smirnov with Sidak. Right: Cumulative distribution plots of the absolute cross-validated single-trial prediction errors obtained using a Bayes classifier (naive approximation for computation efficiency) to decode the single-trial response patterns from the 6 (neuropixels) or 7 (sTeFo 2 P imaging) top ranked units in the simultaneously acquired datasets across mice (cyan), modeled decorrelated datasets (orange) and the chance level distribution associated with our stimulation paradigm (gray).

---

further technical improvements such as multiplexing or larger fields of view (*Clough et al., 2021*; *Demas et al., 2021*; *Ota et al., 2021*; *Weisenburger et al., 2019*) and 3P excitation (*Weisenburger et al., 2017*).

The imaged or electrophysiologically recorded population activity datasets here described support with simultaneous recordings that the single-trial response patterns from subsets of neurons with response dependency to stimulus azimuth (top ranked units) constitute a population code for sound azimuth at the DCIC of awake, passively listening mice (*Figures 4–5*). This finding complements previous studies analyzing pooled, non-simultaneous extracellular recordings from the CNIC of passively listening animals (*Day et al., 2012*; *Day and Delgutte, 2016*; *Day and Delgutte, 2013*), which could have potentially exciting implications, as DCIC represents a higher order relay of the auditory pathway, with respect to CNIC, and receives more profuse descending corticofugal inputs involved in sound location experience-dependent plasticity (*Bajo et al., 2019*; *Bajo et al., 2010*; *Lesicko et al., 2022*; *Winer et al., 2002*). Concretely, we show that sound location coding does indeed occur at DCIC on the single-trial basis, and that this follows a comparable mechanism to the characterized population code at CNIC (*Day and Delgutte, 2013*). However, it remains to be determined if indeed the DCIC network is physiologically capable of Bayesian decoding computations. Interestingly, the small number of DCIC top ranked units necessary to effectively decode stimulus azimuth suggests that sound azimuth information is redundantly distributed across DCIC top ranked units, which points out that mechanisms beyond coding efficiency could be relevant for this population code.

While the decoding error observed from our DCIC datasets obtained in passively listening, untrained mice coincidentally matches the discrimination ability of highly trained, motivated mice (*Lauer et al., 2011*), a relationship between decoding error and psychophysical performance remains to be determined. Interestingly, a primary sensory representations should theoretically be even more precise than the behavioral performance as reported in the visual system (*Stringer et al., 2021*). One possible explanation could be that the population code of sound azimuth at DCIC is likely not a primary representation, as DCIC is reported to be involved in higher order functions including experience dependent plasticity, is influenced by auditory cortex (*Bajo et al., 2019*; *Bajo et al., 2010*; *Bajo and King, 2012*) and is associated to non-auditory processes (*Gruters and Groh, 2012*; *Wong and Borst, 2019*). In this respect, we observed a correlation between the recorded DCIC population activity and snout movements, suggesting that non-auditory processes are likely influencing the recorded responses (*Figure 1—figure supplement 2*). On the other hand, our observations were drawn from unilateral datasets from a single IC. It has been reported that the population responses of a single IC (unilateral recordings from CNIC) can produce a complete representation of sound location throughout the frontal hemifield (*Day and Delgutte, 2016*; *Day and Delgutte, 2013*). Our observations support this notion, showing that unilateral subpopulations of top ranked DCIC units carry enough information to generate a representation of the complete frontal hemifield that is accurate enough to match the discrimination ability of mice (*Lauer et al., 2011*). Discrepancies with respect to unilateral lesion studies (*Day and Delgutte, 2013*; *Jenkins and Masterton, 1982*), suggesting that one IC would carry information about the contralateral hemifield only, could be explained by the loss of commissural interactions between both ICs that contribute to sound location processing (*Orton et al., 2016*). In this respect, we observed that the jRGECO1a-labeled neurons we imaged could project contralaterally (*Figure 1A*, inset), suggesting that the imaged DCIC populations would be

involved in commissural interactions. This is particularly interesting in the context of the opposing two channel model of sound location coding proposed for the mammalian auditory brainstem (*Grothe et al., 2010*; *Park et al., 2004*), which was ruled out for azimuth coding at the rabbit CNIC (*Day and Delgutte, 2013*), but could still be relevant for understanding DCIC population coding of sound location. Thus, we cannot rule out that the actual accuracy of the DCIC population code for sound azimuth, on the single-trial basis, would only be reached if we considered bilateral DCIC recordings. These open questions about the transformations undergone by the neural code for sound location across the relays of the auditory pathway and the influence of bilateral interactions are exciting subjects for future study.

Both our imaging and neuropixels datasets show that the responses from DCIC neurons are noisy, independently of the recording methodology or stimulus (*Figures 4 and 5*, *Figure 1—figure supplement 1*), and that this neuronal noise is often positively correlated across simultaneously recorded pairs of top ranked units, which are involved in the population code of sound azimuth (*Figure 6*). Interestingly, these noise correlations can contribute to reducing the error of the DCIC population code for sound azimuth (*Figure 6*). In low dimensional samples of ~two neurons this could only be justified by concomitant negative signal correlations with positively correlated noise (middle scenario in *Figure 5A*). However, the recorded top ranked units typically showed contralateral sensitivity, which means generally positive signal correlation instead. Nevertheless, in larger samples this relationship between signal and noise correlations seems to be more complex, supporting that these assumptions do not necessarily apply in a complex multidimensional representation like the population code here studied, where empirical determination of the impact of noise correlations are more informative (*Montijn et al., 2016*). Noise correlations have been reported to occur at the mammalian CNIC and have a role in sound categorization (*Sadeghi et al., 2019*), however their influence on the population code for sound location remained unexplored (*Day et al., 2012*; *Day and Delgutte, 2016*; *Day and Delgutte, 2013*). Thus, our data supports that response sensitivity to azimuth might not be the only feature of IC neurons carrying information about stimulus azimuth, as considered in previous studies (*Chase and Young, 2008*; *Chase and Young, 2005*; *Day et al., 2012*; *Day and Delgutte, 2016*; *Day and Delgutte, 2013*), but noise correlations across the population responses could also be a relevant factor. This implication might extend to other brain regions relevant for sound location coding or perhaps also to different sensory modalities.

It is worth mentioning that many discrepancies exist in the labeling and segmentation boundaries of the anatomical subdivisions of IC between the most commonly used mouse brain reference atlases (*Chon et al., 2019*), like the Franklin-Paxinos atlas (*Paxinos and Franklin, 2001*) and the Allen atlas ( atlas.brain-map.org), which can lead to conflicting interpretations of experimental data (*Bjerke et al., 2018*). These differences could arise due to different criteria employed by the expert neuroanatomists to segment anatomical subdivisions (histological stainings or magnetic resonance imaging) different tissue preparation (PFA fixed vs. fresh frozen) or intrinsic variability in the colonies of the animals employed (*Chon et al., 2019*). Here, we adopted the IC segmentation boundaries and labels from the Franklin-Paxinos atlas, due to the extensive previous research employing it and the fact that we followed a compatible tissue preparation procedure for determining the anatomical location of our imaged volumes and electrode tracts (PFA fixed tissue slices). Recent efforts implementing the widely used segmentation and labeling from the Franklin-Paxinos atlas into the Allen common coordinate framework are a step in the right direction to circumvent this issue (*Chon et al., 2019*). Taking this into consideration, we report that DCIC top ranked units scatter across the imaged volume with no specificity in their location across the anatomical axes (*Figure 6*). Interestingly, in vivo imaging studies report that the mouse dorsal IC processes sound frequency information topographically (tonotopy), showing a distinct tonotopic arrangement of neurons across the rostromedial-caudolateral and dorso-ventral axes (*Barnstedt et al., 2015*; *Wong and Borst, 2019*). Beyond the mouse model, a hypothetical functional relationship between the distribution of sound location encoding DCIC neuronal subpopulations and DCIC tonotopic gradients of sound frequency tuning could shed light into how the DCIC subpopulation code of sound azimuth relates to the duplex theory of sound location (*Rayleigh, 1907*). Nevertheless, we observed that not all DCIC azimuth-sensitive units show pure tone sound frequency sensitivity, whereas a fraction of units showing both azimuth and sound frequency sensitivity displayed selectivity for high sound frequencies >10 kHz, making this argument only relevant to perhaps the latter subset of DCIC neurons. Technical improvements realizing larger fields of view to interrogate

the wide-stretching tonotopic DCIC map in simultaneous recordings would be key to further explore this relationship (*Demas et al., 2021*; *Ota et al., 2021*).

In conclusion, our simultaneous recordings from passively listening mice directly support that population response patterns from a subset of DCIC noisy neurons effectively encode sound location on the single-trial basis with an error that matches the discrimination ability of trained mice (*Lauer et al., 2011*). Specifically, we report that azimuth information is redundantly distributed across the subset of DCIC azimuth-sensitive units (top ranked units), which rely not only on response dependency to sound location but also on noise-correlations to accurately encode this information and are randomly scattered across DCIC. An important open question remains about the behavioral relevance of this code and the noise correlations. We hope that this study paves the way to explore this question in detail, together with the wealth of knowledge available about cortico-collicular interactions involved in experience-dependent plasticity in the context of sound localization (*Bajo et al., 2019*; *Bajo et al., 2010*).

## Materials and methods

### Animals and ethics statement

This work was performed in compliance to the European Communities Council Directive (2010/63/EU) to minimize animal pain and discomfort. EMBL's committee for animal welfare and institutional animal care and use (IACUC) approved all experimental procedures under protocol number 2019-04-15RP. Experiments were performed on 7–16 week-old CBA/j mice obtained from Charles River Laboratories and housed in groups of 1–5 in makrolon type 2 L cages on ventilated racks at room temperature and 50% humidity with a 12 hr light cycle. Food and water were available ad libitum. Experimental subjects consisted of 12 mice for imaging and 4 mice for electrophysiological recordings.

### Surgical procedures

#### For microscopy

Cranial window surgeries were performed on 7–8 week-old mice of either sex following procedures published elsewhere (*Boffi et al., 2018*). Briefly, anesthesia consisted of a mixture of 40 µl fentanyl (0.1 mg/ml; Janssen), 160 µl midazolam (5 mg/ml; Hameln), and 60 µl medetomidin (1 mg/ml; Pfizer), dosed in 5 µl/g body weight and injected i.p. Hair removal cream was used to remove the fur over the scalp of anesthetized mice and eye ointment was applied (Bepanthen, Bayer). 1% xylocain (AstraZeneca) was injected as preincisional anesthesia under the scalp. Prepared mice were then placed in a stereotaxic apparatus (David Kopf Instruments, model 963) with their bodies on a heating pad (37 °C). The scalp was surgically removed to expose the dorsal cranium. The periosteum was removed with fine forceps and scissors to prepare the surface for cranial window implantation and stereotaxic delivery of viral vectors. Post-surgical pain relief was provided (Metacam, Boehringer Ingelheim) through s.c. injections (0.1 mg/ml, dosed 10 µl/g body weight).

The $Ca^{2+}$ indicator jRGECO1a (*Dana et al., 2016*) was expressed in IC neurons through transduction with AAV vectors (Addgene #100854-AAV1), which were stereotaxically delivered as follows. A 4 mm diameter circular craniectomy centered over IC (~1 mm posterior to lambda (*34*)) was produced using a dental drill (Microtorque, Harvard Apparatus) with care to avoid bleeding and damage of the dura, which was not removed and left intact. Stereotaxic injections were performed at the right IC (from bregma: –5.2 mm AP, 0.5 mm ML) with pulled glass pipettes lowered to depths of 300, 400, and 500 µm, at a rate of ~4 µl/hr using a 10 ml syringe (to generate pressure) coupled o the glass needle through a silicon tubing via a luer three-way T valve. ~300 nl were injected per site. After injection, the craniectomy was sealed with a round 4 mm coverslip (~170 µm thick, disinfected with 70% ethanol), with a drop of saline between the glass and the dura, and dental acrylic cement (Hager Werken Cyano Fast and Paladur acrylic powder). A head fixation bar was also cemented. The open skin wound was also closed with acrylic cement. At the end of the surgery anesthesia was antagonized with a subcutaneous injection of a mixture of 120 µl sterile saline, 800 µl flumazenil (0.1 mg/ml; Fresenius Kabi), and 60 µl atipamezole (5 mg/ml; Pfizer) dosed in 10 µl/g body weight. Mice were single housed after surgery to minimize the chance of cage mates compromising each others implantations and were allowed to recover for at least 4 weeks before imaging, providing time for $Ca^{2+}$ indicator expression and for the inflammation associated with this surgery to resolve (*Holtmaat et al., 2009*).

## For electrophysiology

Acute craniectomy surgeries were performed on 7–15 week-old mice of either sex following procedures published elsewhere (*Boffi et al., 2018*). Briefly, anesthesia was induced with 5% isoflurane (Baxter) in $O_2$, and maintained at 1.5–2% with a flow rate of 0.4–0.6 LPM. Hair removal cream was used to remove the fur over the scalp, eye ointment was applied (Bepanthen, Bayer) and 1% xylocain (AstraZeneca) was injected under the scalp as preincisional anesthesia. Mice were set in the stereotaxic apparatus with their bodies on a heating pad (37 °C) and their scalp and periosteum removed to expose the dorsal cranium. A custom made head bar was cemented to the exposed cranium, sealing the skin wound, with UV cured dental acrylic to reduce curing time (Loctite 4305 LC). A ~4 mm diameter circular craniectomy centered over IC (~1 mm posterior to lambda *Paxinos and Franklin, 2001*) was produced using a dental drill (Microtorque, Harvard Apparatus) with care to avoid bleeding and damage of the dura. The surface of the brain was kept moist at all times with sterile cortex buffer (mM: NaCl 125, KCl 5, Glucose 10, Hepes 10, $CaCl_2$ 2, $MgCl_2$ 2, pH 7.4). The dura was carefully ripped open at a small site over DCIC for electrode insertion, through an incision made with a sterile G27 needle. The dura was not completely removed and the rest was left intact. Post-surgical pain relief was provided (Metacam, Boehringer Ingelheim) through s.c. injections (0.1 mg/ml, dosed 10 µl/g body weight).

Craniectomy surgeries typically lasted 30 min since the induction of anesthesia, after which the mouse was allowed to recover headfixed at the recording setup (same as for $Ca^{2+}$ imaging, see below). Typically ~5 min after isoflurane removal the mice displayed awake behaviors like whisking, blinking, grooming, and body movements. The craniectomy was kept submerged in a well of cortex buffer throughout the experiment. We began recording at least 40 min after the mouse showed signs of being fully awake. During this period, the neuropixel probe was inserted into the DCIC at 10 µm/s rate using a micromanipulatior (Sutter MPC-385 system, coupled to a Sensapex uMp-NPH Neuropixel 1.0 probe holder). After insertion, neuropixel probes were allowed to settle in the tissue for 10 min. The metal headbar was left exposed to the cortex buffer close to the edge of the craniectomy and was used as reference.

## sTeFo-2P microscopy

We built a bespoke scanned temporal focusing (sTeFo) 2 photon microscope for fast volumetric in vivo $Ca^{2+}$ imaging of large IC neuronal populations from awake mice. Technical details and detailed working principle are extensively described elsewhere (*Prevedel et al., 2016*). The main difference with respect to the microscope design published by *Prevedel et al., 2016* was a higher repetition laser (10 MHz, FemtoTrain, Spectra Physics). Laser power while scanning was kept below 191 mW, measured after the objective, to avoid heating the brain tissue excessively (*Prevedel et al., 2016*). 470 $\mu m^2$ fields of view were imaged at 128 $px^2$ resolution and spaced in 15 µm z steps to cover 570 µm in depth (38 z steps) at a volume rate of 4.01 Hz. Artifacts produced by objective piezo z drive flyback were excluded from analysis.

In a typical imaging session, cranial window implanted mice were briefly (<1 min) anesthetized with 5% isoflurane in $O_2$ for quick head fixation at a custom stage, slightly restraining their bodies inside a 5 cm diameter acrylic tube, and positioned under our custom 2P microscope. Mice fully recovered from the brief isoflurane anesthesia, showing a clear blinking reflex, whisking and sniffing behaviors and normal body posture and movements, immediately after head fixation. In our experimental conditions, mice were imaged in sessions of up to 25 min since beyond this time we started observing some signs of distress or discomfort. Thus, we avoided longer recording times at the expense of collecting larger trial numbers, in strong adherence of animal welfare and ethics policy. A pilot group of mice were habituated to the head fixed condition in daily 20 min sessions for 3 days; however, we did not observe a marked contrast in the behavior of habituated versus unhabituated mice beyond our relatively short 25-min imaging sessions. In consequence imaging sessions never surpassed a maximum of 25 min, after which the mouse was returned to its home cage. Typically, mice were imaged a total of 2–11 times (sessions), one to three times a week. Datasets here analyzed and reported come from the imaging session in which we observed maximal calcium sensor signal (peak AAV expression) and maximum number of detected units. Volumetric imaging data was visualized and rendered using FIJI (*Schindelin et al., 2012*). Motion correction was performed using NoRMCorre (*Pnevmatikakis and Giovannucci, 2017*) on individual imaging planes from the volumetric datasets. We used the CaImAn

package (*Giovannucci et al., 2019*) for automatic ROI segmentation through constrained non negative matrix factorization and selected ROIs (Units) showing clear Ca transients consistent with neuronal activity, and IC neuron somatic shape and size (*Schofield and Beebe, 2019*). jRGECO1a signal from the segmented ROIs was extracted, denoised and neuropil corrected from each individual imaging plane using CalmAn (*Giovannucci et al., 2019*). Spike probabilities were estimated using MLSpike (*Deneux et al., 2016*).

## Extracellular electrophysiological recordings

We performed acute extracellular single-unit recordings using the Neuropixles (*Jun et al., 2017*) 1.0 system (PRB_1_4_0480_1, PXIe_1000, HST_1000, IMEC). Reference and ground pads on the band connector of the probe were bridged and grounded. The probe shank was coated before recordings by dipping 10 times every 5 s in DiI (V22885, Thermo Fisher) for post hoc histological assessment of electrode tracks. Recordings were performed using SpikeGLX software (https://billkarsh.github.io/SpikeGLX/). Mice were recorded for ~25 min after successful electrode insertion (see surgical procedures), after which they were placed in an empty cage, and their behavior monitored for 10–15 min. We only considered recordings from mice that displayed normal behaviors (locomotion, exploration, sniffing, whisking), with no clear signs of pain or discomfort (freezing, shivering, raised fur) at this stage. Mice were sacrificed after this with $CO_2$ for transcardial perfusion and tissue preparation to trace electrode tracks (see below). After retrieval, neuropixels probes were cleaned through immersion in 1% Tergazime (Z742918, Merck) for some minutes, followed by washes in distilled water and reused multiple times. Recordings were preprocessed for spike sorting with CatGT (https://billkarsh.github.io/SpikeGLX/#catgt) for high pass filtering, common average referencing of the demultiplexed channels and file concatenation. Spike sorting with drift correction was performed using kilosort 2.5 (*Pachitariu et al., 2020*). The output of kilosort was manually curated using Phy (*Rossant, 2021*, https://github.com/cortex-lab/phy), selecting only single-units with somatic AP waveforms with good waveform consistency (amplitude) across spikes, showing few or no refractory period violations (2ms refractory period violation time window) in their correlogram plots and firing a minimum of 150 times in 20 min of recording. Stimulus times collected as digital channel inputs and recorded clock signals from the neuropixles headstage were temporaly aligned to a reference clock with Tprime (https://billkarsh.github.io/SpikeGLX/#tprime).

## Sound stimulation

A custom sound-attenuating chamber was incorporated to the microscope/electrophysiology setup, allowing only the objective or cables to access the inside, to attenuate room and microscope scanner noise reaching the mouse. All possible sound reflecting surfaces were lined with 1-cm or 0.5-cm-thick foam to preserve free-field conditions. SPL was measured using a free-field prepolarized measurement microphone (PCB Piezotronics, 378C01, with signal conditioner 482A21) placed on the setup stage in the position of the head of the mouse. For imaging experiments, background R.M.S. SPL measurements during imaging without sound stimuli delivery was (20–40 kHz) 44.83 dB (79.5 dB total SPL summed across the band). Sound stimulus (20–40 kHz) R.M.S. SPL during imaging was 56.83 dB (96.2 dB total SPL summed across the band), ensuring that SPL was at least 10 dB above background SPL for sound stimuli to be salient. Background R.M.S. SPL during imaging across the mouse hearing range (2.5–80 kHz) was 44.53 dB.

Electrophysiological experiments were performed at the same setup, without the microscope running. For broadband stimuli applied in our electrophysiological experiments, background R.M.S. SPL (15–50 kHz) was 35.96 dB (89.9 dB total SPL summed across the band). Broadband stimulus (15–50 kHz) R.M.S. SPL was 62.5 dB (102 dB total SPL summed across the band). Pure tone stimuli consisted on 4–48 kHz tones, making 3.5 octaves split into steps of 4. Peak SPL of the pure tone stimuli was calibrated to 65±3 dB. R.M.S. background noise across the 4–48 kHz band was 35.1 dB. Background R.M.S. SPL across the mouse hearing range (2.5–80 kHz) was 34.68 dB.

Broadband sound stimuli consisted of band-passed frozen noise. Stimulus duration was 500ms for imaging experiments and 200ms for electrophysiological experiments, including 10ms up and down ramps. Sound stimuli were delivered through an electrostatic speaker (Tucker Davis Technologies, ES1 coupled to a ED1 speaker driver) mounted on an arm at an 8.5 cm distance away from the head of the mouse, which was mounted on a NEMA17 bipolar stepper motor (26 Ncm, 1.8 deg/step) placed

under the head of the mouse to position the speaker around the frontal hemifield. The motor was controlled using an Arduino UNO microcontroller, running GRBL v0.9 (*Skogsrud and Jeon, 2016*) and coupled with the Arduino CNC Shield V3.10 (https://blog.protoneer.co.nz/arduino-cnc-controller/) running A4988 motor drivers. To control and program speaker positioning, we used the universal G-code sender (https://winder.github.io/ugs_website/). Synchronization of speaker positioning and sound stimulus delivery to data acquisition was done through a second Arduino UNO board receiving a scan start TTL from the microscope to trigger stimulation programs or by recording the sound delivery TTL signal with a DAQ board (National Instruments, PXIe-6341) through a digital channel in SpikeGLX. Pure tone stimuli were presented with the speaker parked at 0° azimuth, in front of the mouse.

Audio stimuli (including 10ms up and down ramps) were synthesized using MATLAB (0.5 MHz sampling rate). Sound delivery was performed using a DAQ board (National Instruments, PXIe-6341) interfaced to MATLAB using the DAQmx wrapper from scanimage (https://vidriotechnologies.com/scanimage/).

A typical stimulation protocol consisted of a series of presentations of a sound stimulus in which azimuth angles varied randomly from 13 different frontal positions (frontal 180° split into 15° steps). For imaging experiments, stimuli were presented every 5 s and each azimuth angle was presented on average 14 times per session. Minimum number of same-azimuth trials collected was 8. For electrophysiological experiments, firstly pure tone stimuli were applied every 2 s in pseudo-random order to obtain 10 trial repetitions (14 sound frequencies, 140 trials), and after that broadband stimulus azimuth trials were applied every 3 s (to allow the motor to complete speaker movements) in pseudo-random order to obtain 20 trial repetitions (13 azimuths, 260 trials).

## Histology

At the end of the experiment mice were transcardially perfused with phosphate-buffered saline (PBS) followed by 4% paraformaldehyde at room temperature to fixate the brain. The perfused brain was dissected and post-fixed in 4% paraformaldehyde for 24 hr at 4 °C. After post-fixation, brains were washed with PBS saline and stored at 4 °C. 100-μm-thick free-floating vibratome (Leica VT1200) coronal sections were cut at room temperature, mounted on slides using Vectashield with DAPI (H-1200–10, Vector Laboratories) and imaged in a NIKON Ti-E epifluorescence microscope equipped with a LUMENCOR SPECTRA X module for illumination, standard DAPI and mCherry filter cubes, and a pco.edge 4.2 CL sCMOS camera; using a CFI P-Apo 4 x Lambda/ 0.20/20,00 objective. Images of brain slices were aligned to the mouse brain reference atlas manually following anatomical landmarks.

## Face movement videography

Videos from one side of the mouse face, ipsilateral to the imaged IC, were recorded with an IR sensitive camera equipped with a CMOS OV2710 sensor, IR LEDs for illumination, a 25mm M12 objective (ELP, USBFHD05MT-KL36IR), using a sampling rate of 30 fps at 720 p resolution. Video acquisition was synchronized to the microscope acquisition start. Video image analysis based on intensity change between successive frames was performed following a similar approach as described by *Wong and Borst, 2019* using FIJI (*Schindelin et al., 2012*) and custom MATLAB scripts. For correlations between face movements and imaged neuronal activity, the Pearson correlation coefficient was used.

## Statistical analyses

Each mouse was considered a replicate. Datasets obtained in one imaging/recording session from each mouse, consisting of ~200 azimuth trials were analyzed. We did not perform any form of data augmentation through resampling with restitution of the recorded responses to virtually increase the number of trials (*Day and Delgutte, 2013*). Statistical analyses were performed using MATLAB (2023a,b) functions and custom scripts. For hypothesis testing, a p-value <0.05 was considered significant. Sidak's correction of p values was used for multiple comparisons. nS/N ratios were evaluated as the inverse of the coefficient of variation of the recorded responses of each cell across azimuth trials.

For our imaging data, since many units failed to respond across many trials, we did not perform baseline spike probability subtraction as this frequently led to negative spike probabilities that were hard to interpret. For our neuropixels experiments, we generally did not perform baseline firing subtraction from the responses as the average baseline firing in the 200ms before stimuli was quite

low and close to 0 (*Figure 1—figure supplement 1F*, *Figure 5A*). Nevertheless, we did subtract baseline firing for the determination of the nS/N (*Figure 5C*), as the on going activity varied extensively across the recordings (in the absence of sound stimulation, *Figure 1I*) and this could lead to nS/N values that were artificially high. On going activity was estimated as the spike count determined in 200ms time windows during the inter trial periods (2.8 s, no sound stimulation), 1 s before each stimulation trial. 'Baseline on-going firing' was determined as the spike count 200ms previous to that. For fairness of comparisons, we also subtracted baseline firing determined during the 200ms pre response for nS/N determination from the sound stimulated condition.

One sided Kruskal-Wallis tests were performed to determine median response tuning to stimulus azimuth, with 12 degrees of freedom (13 tested azimuths). Two sample Kolmogorov-Smirnov and Wilcoxon rank sum tests were two sided. To find units with single-trial responses that were statistically dependent on stimulus azimuth (feature selection) we performed $\chi^2$ independency tests using the function fscchi2 from the statistics and machine learning toolbox from MATLAB with default parameters, which binned the responses into 10 bins. Cross validated sorting of noise correlation matrices was obtained by performing hierarchical clustering on half of the trials recorded based on pairwise neuronal noise correlations (using the Kendall tau as a non parametric correlation coefficient) as a distance metric, and plotting the pairwise correlation coefficient matrix registered for the other half of the trials, sorting the units based on the clustering. IC population response classification (decoding) was performed using the naive Bayes classifier algorithm implemented in MATLAB, using the fitcnb function from the statistics and machine learning toolbox. For the latter function, model fit involved Bayesian hyperparameter optimization for all eligible parameters. This MATLAB function performs 3 steps: 1) Estimation of the densities of the units' responses (predictors) within each class (stimulus azimuth) through kernel smoothing density estimation. Uniform, Epanechnikov, Gaussian or triangular smoothing kernels and Kernel smoothing window width were set through hyperparameter optimization. 2) Use the Bayes rule to estimate the posterior probability P^ for all possible azimuth classes k = 1, ... , 13 (–90° to 90° in 15° steps) as:

$$\hat{P}(Y = K \mid X_1...X_P) = \frac{\pi(Y = k) \prod\limits_{j=1}^{p} P(X_j \mid Y = k)}{\sum\limits_{k=1}^{13} \pi(Y = k) \prod\limits_{j=i}^{p} P(X_j \mid Y = k)},$$

where *Y* is the possible azimuth class being evaluated, $X_1,...,X_p$ are the simultaneously acquired responses from the units in a population of size *p* during a single azimuth trial, $\pi(Y = k)$ is the prior probability that the azimuth class is *k*, determined as the relative frequencies of each azimuth class in the training dataset. (3) Determine the azimuth class of a single-trial response pattern as the class with maximum posterior probability (maximum a posteriori decision rule).

For dimensionality reduction, feature selection and classification (decoding) a 'leave one out' cross-validation strategy was implemented for the whole process, in which dimensionality reduction (PCA) or feature selection (top ranked cell selection, if applicable) was performed and then the models were fitted to the population responses collected in all trials except one (for each imaging session) and the one population response left out was decoded as is or using the selected features (best cells) or using the PCA loadings matrix to calculate PC scores, and fitted model. This procedure was iterated until all population responses simultaneously recorded in an experiment (mouse) were decoded.

## Materials and correspondence

Datasets and code supporting this study are available from https://www.ebi.ac.uk/biostudies/bioimages/studies/S-BIAD1064.

## Acknowledgements

We acknowledge Lina Streich and Ling Wang for their contribution to microscope assembly and setup, the mechanical and electronics workshops and the laboratory animal resource facility of EMBL Heidelberg for technical assistance. Evan Harrell for input on motorized speaker design and Jacques Bourg for input on IC cranial window surgeries. Peter Rupprecht and Alejandro Tlaie-Boria for helpful comments on the manuscript. JCB acknowledges supporting fellowships from the EMBL

Interdisciplinary Postdoc (EIPOD) Programme under Marie Skłodowska Curie Cofund Actions MSCA-COFUND-FP (664726). This work was funded by the European Molecular Biology Laboratory (EMBL) and the Deutsche Forschungsgemeinschaft (DFG, German Research Foundation) – project 458898724 awarded to RP.

## Additional information

### Competing interests

Brice Bathellier: Reviewing editor, *eLife*. The other authors declare that no competing interests exist.

### Funding

| Funder | Grant reference number | Author |
|---|---|---|
| Deutsche Forschungsgemeinschaft | 458898724 | Robert Prevedel |
| Fondation Pour l'Audition | FPA IDA02, RD-2023-1 | Brice Bathellier |
| European Molecular Biology Laboratory | | Robert Prevedel Hiroki Asari |
| Marie Skłodowska-Curie Actions | EMBL Interdisciplinary Postdoc (EIPOD) Programme MSCA-COFUND-FP 664726 | Juan Carlos Boffi |

The funders had no role in study design, data collection and interpretation, or the decision to submit the work for publication.

### Author contributions

Juan Carlos Boffi, Conceptualization, Data curation, Software, Formal analysis, Funding acquisition, Validation, Investigation, Visualization, Methodology, Writing - original draft, Project administration, Writing – review and editing; Brice Bathellier, Conceptualization, Resources, Formal analysis, Supervision, Writing – review and editing; Hiroki Asari, Conceptualization, Resources, Software, Formal analysis, Supervision, Writing – review and editing; Robert Prevedel, Conceptualization, Resources, Formal analysis, Supervision, Funding acquisition, Project administration, Writing – review and editing

### Author ORCIDs

Juan Carlos Boffi https://orcid.org/0000-0002-0116-6892
Brice Bathellier https://orcid.org/0000-0001-9211-1960
Hiroki Asari http://orcid.org/0000-0003-3396-1935
Robert Prevedel http://orcid.org/0000-0003-3366-4703

### Ethics

This work was performed in compliance to the European Communities Council Directive (2010/63/EU) to minimize animal pain and discomfort. EMBL's committee for animal welfare and institutional animal care and use (IACUC) approved all experimental procedures under protocol number 2019-04-15RP.

Reviewer #1 (Public review): https://doi.org/10.7554/eLife.97598.4.sa1
Reviewer #2 (Public review): https://doi.org/10.7554/eLife.97598.4.sa2
Reviewer #3 (Public review): https://doi.org/10.7554/eLife.97598.4.sa3
Author response https://doi.org/10.7554/eLife.97598.4.sa4

## Additional files

### Supplementary files
• MDAR checklist

## Data availability

Datasets and code supporting this study are available from https://www.ebi.ac.uk/biostudies/bioimages/studies/S-BIAD1064.

The following dataset was generated:

| Author(s) | Year | Dataset title | Dataset URL | Database and Identifier |
|---|---|---|---|---|
| Boffi JCA, Prevedel R, Asari H, Bathellier B | 2024 | Data supporting "Noisy neuronal populations effectively encode sound localization in the dorsal inferior colliculus of awake mice" | https://doi.org/10.6019/S-BIAD1064 | BIAD1064, 10.6019/S-BIAD1064 |

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
