## [Editor Report · eLife Assessment]

The paper reports the **important** discovery that the mouse dorsal inferior colliculus, an auditory midbrain area, encodes sound location. The evidence supporting the claims is **solid**, being supported by both optical and electrophysiological recordings. The observations described should be of interest to auditory researchers studying the neural mechanisms of sound localization and the role of noise correlations in population coding.

---

## [Referee Report · Reviewer #1 (Public review)]

Summary:

In this study, the authors address whether the dorsal nucleus of the inferior colliculus (DCIC) in mice encodes sound source location within the front horizontal plane (i.e., azimuth). They do this using volumetric two-photon Ca2+ imaging and high-density silicon probes (Neuropixels) to collect single-unit data. Such recordings are beneficial because they allow large populations of simultaneous neural data to be collected. Their main results and the claims about those results are the following:

(1) DCIC single-unit responses have high trial-to-trial variability (i.e., neural noise);

(2) approximately 32% to 40% of DCIC single units have responses that are sensitive to sound source azimuth;

(3) single-trial population responses (i.e., the joint response across all sampled single units in an animal) encode sound source azimuth "effectively" (as stated in the title) in that localization decoding error matches average mouse discrimination thresholds;

(4) DCIC can encode sound source azimuth in a similar format to that in the central nucleus of the inferior colliculus (as stated in the Abstract);

(5) evidence of noise correlation between pairs of neurons exists;

and (6) noise correlations between responses of neurons help reduce population decoding error.

While simultaneous recordings are not necessary to demonstrate results #1, #2, and #4, they are necessary to demonstrate results #3, #5, and #6.

Strengths:

- Important research question to all researchers interested in sensory coding in the nervous system.

- State-of-the-art data collection: volumetric two-photon Ca2+ imaging and extracellular recording using high-density probes. Large neuronal data sets.

- Confirmation of imaging results (lower temporal resolution) with more traditional microelectrode results (higher temporal resolution).

- Clear and appropriate explanation of surgical and electrophysiological methods. I cannot comment on the appropriateness of the imaging methods.

Strength of evidence for the claims of the study:

(1) DCIC single-unit responses have high trial-to-trial variability -

The authors' data clearly shows this.

(2) Approximately 32% to 40% of DCIC single units have responses that are sensitive to sound source azimuth -

The sensitivity of each neuron's response to sound source azimuth was tested with a Kruskal-Wallis test, which is appropriate since response distributions were not normal. Using this statistical test, only 8% of neurons (median for imaging data) were found to be sensitive to azimuth, and the authors noted this was not significantly different than the false positive rate. The Kruskal-Wallis test was not reported for electrophysiological data. The authors suggested that low numbers of azimuth-sensitive units resulting from the statistical analysis may be due to the combination of high neural noise and a relatively low number of trials, which would reduce the statistical power of the test. This is likely true and highlights a weakness in the experimental design (i.e., a relatively small number of trials). The authors went on to perform a second test of azimuth sensitivity-a chi-squared test-and found 32% (imaging) and 40% (e-phys) of single units to have statistically significant sensitivity. However, the use of a chi-squared test is questionable because it is meant to be used between two categorical variables, and neural response had to be binned before applying the test.

(3) Single-trial population responses encode sound source azimuth "effectively" in that localization decoding error matches average mouse discrimination thresholds -

If only one neuron in a population had responses that were sensitive to azimuth, we would expect that decoding azimuth from observation of that one neuron's response would perform better than chance. By observing the responses of more than one neuron (if more than one were sensitive to azimuth), we would expect performance to increase. The authors found that decoding from the whole population response was no better than chance. They argue (reasonably) that this is because of overfitting of the decoder model-too few trials were used to fit too many parameters-and provide evidence from decoding combined with principal components analysis which suggests that overfitting is occurring. What is troubling is the performance of the decoder when using only a handful of "top-ranked" neurons (in terms of azimuth sensitivity) (Fig. 4F and G). Decoder performance seems to increase when going from one to two neurons, then decreases when going from two to three neurons, and doesn't get much better for more neurons than for one neuron alone. It seems likely there is more information about azimuth in the population response, but decoder performance is not able to capture it because spike count distributions in the decoder model are not being accurately estimated due to too few stimulus trials (14, on average). In other words, it seems likely that decoder performance is underestimating the ability of the DCIC population to encode sound source azimuth.

To get a sense of how effective a neural population is at coding a particular stimulus parameter, it is useful to compare population decoder performance to psychophysical performance. Unfortunately, mouse behavioral localization data do not exist. Instead, the authors compare decoder error to mouse left-right discrimination thresholds published previously by a different lab. However, this comparison is inappropriate because the decoder and the mice were performing different perceptual tasks. The decoder is classifying sound sources to 1 of 13 locations from left to right, whereas the mice were discriminating between left or right sources centered around zero degrees. The errors in these two tasks represent different things. The two data sets may potentially be more accurately compared by extracting information from the confusion matrices of population decoder performance. For example, when the stimulus was at -30 deg, how often did the decoder classify the stimulus to a lefthand azimuth? Likewise, when the stimulus was +30 deg, how often did the decoder classify the stimulus to a righthand azimuth?

(4) DCIC can encode sound source azimuth in a similar format to that in the central nucleus of the inferior colliculus -

It is unclear what exactly the authors mean by this statement in the Abstract. There are major differences in the encoding of azimuth between the two neighboring brain areas: a large majority of neurons in the CNIC are sensitive to azimuth (and strongly so), whereas the present study shows a minority of azimuth-sensitive neurons in the DCIC. Furthermore, CNIC neurons fire reliably to sound stimuli (low neural noise), whereas the present study shows that DCIC neurons fire more erratically (high neural noise).

(5) Evidence of noise correlation between pairs of neurons exists -

The authors' data and analyses seem appropriate and sufficient to justify this claim.

(6) Noise correlations between responses of neurons help reduce population decoding error -

The authors show convincing analysis that performance of their decoder increased when simultaneously measured responses were tested (which include noise correlation) than when scrambled-trial responses were tested (eliminating noise correlation). This makes it seem likely that noise correlation in the responses improved decoder performance. The authors mention that the naïve Bayesian classifier was used as their decoder for computational efficiency, presumably because it assumes no noise correlation and, therefore, assumes responses of individual neurons are independent of each other across trials to the same stimulus. The use of a decoder that assumes independence seems key here in testing the hypothesis that noise correlation contains information about sound source azimuth. The logic of using this decoder could be more clearly spelled out to the reader. For example, if the null hypothesis is that noise correlations do not carry azimuth information, then a decoder that assumes independence should perform the same whether population responses are simultaneous or scrambled. The authors' analysis showing a difference in performance between these two cases provides evidence against this null hypothesis.

Minor weakness:

- Most studies of neural encoding of sound source azimuth are done in a noise-free environment, but the experimental setup in the present study had substantial background noise. This complicates comparison of the azimuth tuning results in this study to those of other studies. One is left wondering if azimuth sensitivity would have been greater in the absence of background noise, particularly for the imaging data where the signal was only about 12 dB above the noise.

---

## [Referee Report · Reviewer #2 (Public review)]

In the present study, Boffi et al. investigate the manner in which the dorsal cortex of the of the inferior colliculus (DCIC), an auditory midbrain area, encodes sound location azimuth in awake, passively listening mice. By employing volumetric calcium imaging (scanned temporal focusing or s-TeFo), complemented with high-density electrode electrophysiological recordings (neuropixels probes), they show that sound-evoked responses are exquisitely noisy, with only a small portion of neurons (units) exhibiting spatial sensitivity. Nevertheless, a naïve Bayesian classifier was able to predict the presented azimuth based on the responses from small populations of these spatially sensitive units. A portion of the spatial information was provided by correlated trial-to-trial response variability between individual units (noise correlations). The study presents a novel characterization of spatial auditory coding in a non-canonical structure, representing a noteworthy contribution specifically to the auditory field and generally to systems neuroscience, due to its implementation of state-of-the-art techniques in an experimentally challenging brain region. However, nuances in the calcium imaging dataset and the naïve Bayesian classifier warrant caution when interpreting some of the results.

Strengths:

The primary strength of the study lies in its methodological achievements, which allowed the authors to collect a comprehensive and novel dataset. While the DCIC is a dorsal structure, it extends up to a millimetre in depth, making it optically challenging to access in its entirety. It is also more highly myelinated and vascularised compared to e.g., the cerebral cortex, compounding the problem. The authors successfully overcame these challenges and present an impressive volumetric calcium imaging dataset. Furthermore, they corroborated this dataset with electrophysiological recordings, which produced overlapping results. This methodological combination ameliorates the natural concerns that arise from inferring neuronal activity from calcium signals alone, which are in essence an indirect measurement thereof.

Another strength of the study is its interdisciplinary relevance. For the auditory field, it represents a significant contribution to the question of how auditory space is represented in the mammalian brain. "Space" per se is not mapped onto the basilar membrane of the cochlea and must be computed entirely within the brain. For azimuth, this requires the comparison between miniscule differences between the timing and intensity of sounds arriving at each ear. It is now generally thought that azimuth is initially encoded in two, opposing hemispheric channels, but the extent to which this initial arrangement is maintained throughout the auditory system remains an open question. The authors observe only a slight contralateral bias in their data, suggesting that sound source azimuth in the DCIC is encoded in a more nuanced manner compared to earlier processing stages of the auditory hindbrain. This is interesting because it is also known to be an auditory structure to receive more descending inputs from the cortex.

Systems neuroscience continues to strive for the perfection of imaging novel, less accessible brain regions. Volumetric calcium imaging is a promising emerging technique, allowing the simultaneous measurement of large populations of neurons in three dimensions. But this necessitates corroboration with other methods, such as electrophysiological recordings, which the authors achieve. The dataset moreover highlights the distinctive characteristics of neuronal auditory representations in the brain. Its signals can be exceptionally sparse and noisy, which provide an additional layer of complexity in the processing and analysis of such datasets. This will undoubtedly be useful for future studies of other less accessible structures with sparse responsiveness.

Weaknesses:

Although the primary finding that small populations of neurons carry enough spatial information for a naïve Bayesian classifier to reasonably decode the presented stimulus is not called into question, certain idiosyncrasies, in particular the calcium imaging dataset and model, complicate specific interpretations of the model output, and the readership is urged to interpret these aspects of the study's conclusions with caution.

I remain in favour of volumetric calcium imaging as a suitable technique for the study, but the presently constrained spatial resolution is insufficient to unequivocally identify regions of interest as cell bodies (and are instead referred to as "units" akin to those of electrophysiological recordings). It remains possible that the imaging set is inadvertently influenced by non-somatic structures (including neuropil), which could report neuronal activity differently than cell bodies. Due to the lack of a comprehensive ground-truth comparison in this regard (which to my knowledge is impossible to achieve with current technology), it is difficult to imagine how many informative such units might have been missed because their signals were influenced by spurious, non-somatic signals, which could have subsequently misled the models. The authors reference the original Nature Methods article (Prevedel et al., 2016) throughout the manuscript, presumably in order to avoid having to repeat previously published experimental metrics. But the DCIC is neither the cortex nor hippocampus (for which the method was originally developed) and may not have the same light scattering properties (not to mention neuronal noise levels). Although the corroborative electrophysiology data largely alleviates these concerns for this particular study, the readership should be cognisant of such caveats, in particular those who are interested in implementing the technique for their own research.

A related technical limitation of the calcium imaging dataset is the relatively low number of trials (14) given the inherently high level of noise (both neuronal and imaging). Volumetric calcium imaging, while offering a uniquely expansive field of view, requires relatively high average excitation laser power (in this case nearly 200 mW), a level of exposure the authors may have wanted to minimise by maintaining a low number of repetitions, but I yield to them to explain. Calcium imaging is also inherently slow, requiring relatively long inter-stimulus intervals (in this case 5 s). This unfortunately renders any model designed to predict a stimulus (in this case sound azimuth) from particularly noisy population neuronal data like these as highly prone to overfitting, to which the authors correctly admit after a model trained on the entire raw dataset failed to perform significantly above chance level. This prompted them to feed the model only with data from neurons with the highest spatial sensitivity. This ultimately produced reasonable performance (and was implemented throughout the rest of the study), but it remains possible that if the model was fed with more repetitions of imaging data, its performance would have been more stable across the number of units used to train it. (All models trained with imaging data eventually failed to converge.) However, I also see these limitations as an opportunity to improve the technology further, which I reiterate will be generally important for volume imaging of other sparse or noisy calcium signals in the brain.

Transitioning to the naïve Bayesian classifier itself, I first openly ask the authors to justify their choice of this specific model. There are countless types of classifiers for these data, each with their own pros and cons. Did they actually try other models (such as support vector machines), which ultimately failed? If so, these negative results (even if mentioned en passant) would be extremely valuable to the community, in my view. I ask this specifically because different methods assume correspondingly different statistical properties of the input data, and to my knowledge naïve Bayesian classifiers assume that predictors (neuronal responses) are assumed to be independent within a class (azimuth). As the authors show that noise correlations are informative in predicting azimuth, I wonder why they chose a model that doesn't take advantage of these statistical regularities. It could be because of technical considerations (they mention computing efficiency), but I am left generally uncertain about the specific logic that was used to guide the authors through their analytical journey.

In a revised version of the manuscript, the authors indeed justify their choice of the naïve Bayesian classifier as a conservative approach (not taking into account noise correlations), which could only improve with other models (that do). They even tested various other commonly used models, such as support vector machines and k-nearest neighbours, to name a few, but do not report these efforts in the main manuscript. Interestingly, these models, which I supposed would perform better in fact did not overall - a finding that I have no way of interpreting but nevertheless find interesting.

That aside, there remain other peculiarities in model performance that warrant further investigation. For example, what spurious features (or lack of informative features) in these additional units prevented the models of imaging data from converging? In an orthogonal question, did the most spatially sensitive units share any detectable tuning features? A different model trained with electrophysiology data in contrast did not collapse in the range of top-ranked units plotted. Did this model collapse at some point after adding enough units, and how well did that correlate with the model for the imaging data? How well did the form (and diversity) of the spatial tuning functions as recorded with electrophysiology resemble their calcium imaging counterparts? These fundamental questions could be addressed with more basic, but transparent analyses of the data (e.g., the diversity of spatial tuning functions of their recorded units across the population). Even if the model extracts features that are not obvious to the human eye in traditional visualisations, I would still find this interesting.

Although these questions were not specifically addressed in the revised version of the manuscript, I also admit that I did not indent do assert that these should necessarily fall within the scope of the present study. I rather posed them as hypothetical directions one could pursue in future studies. Finally, further concerns I had with statements regarding the physiological meaning of the findings have been ameliorated by nicely modified statements, thus bringing transparency to the readership, which I appreciate.

In summary, the present study represents a significant body of work that contributes substantially to the field of spatial auditory coding and systems neuroscience. However, limitations of the imaging dataset and model as applied in the study muddles concrete conclusions about how the DCIC precisely encodes sound source azimuth and even more so to sound localisation in a behaving animal. Nevertheless, it presents a novel and unique dataset, which, regardless of secondary interpretation, corroborates the general notion that auditory space is encoded in an extraordinarily complex manner in the mammalian brain.

---

## [Referee Report · Reviewer #3 (Public review)]

Summary:

Boffi and colleagues sought to quantify the single-trial, azimuthal information in the dorsal cortex of the inferior colliculus (DCIC), a relatively understudied subnucleus of the auditory midbrain. They accomplished this by using two complementary recording methods while mice passively listened to sounds at different locations: calcium imaging that recorded large neuronal populations but with poor temporal precision and multi-contact electrode arrays that recorded smaller neuronal populations with exact temporal precision. DCIC neurons respond variably, with inconsistent activity to sound onset and complex azimuthal tuning. Some of this variably was explained by ongoing head movements. The authors used a naïve Bayes decoder to probe the azimuthal information contained in the response of DCIC neurons on single trials. The decoder failed to classify sound location better than chance when using the raw population responses but performed significantly better than chance when using the top principal components of the population. Units with the most azimuthal tuning were distributed throughout the DCIC, possessed contralateral bias, and positively correlated responses. Interestingly, inter-trial shuffling decreased decoding performance, indicating that noise correlations contributed to decoder performance. Overall, Boffi and colleagues, quantified the azimuthal information available in the DCIC while mice passively listened to sounds, a first step in evaluating if and how the DCIC could contribute to sound localization.

Strengths:

The authors should be commended for collection of this dataset. When done in isolation (which is typical), calcium imaging and linear array recordings have intrinsic weaknesses. However, those weaknesses are alleviated when done in conjunction - especially when the data is consistent. This data set is extremely rich and will be of use for those interested in auditory midbrain responses to variable sound locations, correlations with head movements, and neural coding.

The DCIC neural responses are complex with variable responses to sound onset, complex azimuthal tuning and large inter-sound interval responses. Nonetheless, the authors do a decent job in wrangling these complex responses: finding non-canonical ways of determining dependence on azimuth and using interpretable decoders to extract information from the population.

Weaknesses:

The decoding results are a bit strange, likely because the population response is quite noisy on any given trial. Raw population responses failed to provide sufficient information concerning azimuth for significant decoding. Importantly, the decoder performed better than chance when certain principal components or top ranked units contributed but did not saturate with the addition of components or top ranked units. So, although there is azimuthal information in the recorded DCIC populations - azimuthal information appears somewhat difficult to extract.

Although necessary given the challenges associated with sampling many conditions with technically difficult recording methods, the limited number of stimulus repeats precludes interpretable characterization of the heterogeneity across the population. Nevertheless, the dataset is public so those interested can explore the diversity of the responses.

The observations from Boffi and colleagues raises the question: what drives neurons in the DCIC to respond? Sound azimuth appears to be a small aspect of the DCIC response. For example, the first 20 principal components which explain roughly 80% of the response variance are insufficient input for the decoder to predict sound azimuth above chance. Furthermore, snout and ear movements correlate with the population response in the DCIC (the ear movements are particularly peculiar given they seem to predict sound presentation). Other movements may be of particular interest to control for (e.g. eye movements are known to interact with IC responses in the primate). These observations, along with reported variance to sound onsets and inter-sound intervals, question the impact of azimuthal information emerging from DCIC responses. This is certainly out of scope for any one singular study to answer, but, hopefully, future work will elucidate the dominant signals in the DCIC population. It may be intuitive that engagement in a sound localization task may push azimuthal signals to the forefront of DCIC response, but azimuthal information could also easily be overtaken by other signals (e.g. movement, learning).

Boffi and colleagues set out to parse the azimuthal information available in the DCIC on a single trial. They largely accomplish this goal and are able to extract this information when allowing the units that contain more information about sound location to contribute to their decoding (e.g., through PCA or decoding on their activity specifically). Interestingly, they also found that positive noise correlations between units with similar azimuthal preferences facilitate this decoding - which is unusual given that this is typically thought to limit information. The dataset will be of value to those interested in the DCIC and to anyone interested in the role of noise correlations in population coding. Although this work is first step into parsing the information available in the DCIC, it remains difficult to interpret if/how this azimuthal information is used in localization behaviors of engaged mice.

---

## [Author Response]

The following is the authors’ response to the previous reviews.

**Recommendations for the authors:**

**Reviewer #2 (Recommendations for the authors):**
I appreciate the efforts the authors made to clarify and justify their statements and methodology, respectively. I additionally appreciate the efforts they made to provide me with detailed information - including figures - to aid my comprehension. However, there are two things I nevertheless recommend the authors to include in the main manuscript.(1) Statement about animal wellbeing: The authors state that they were constrained in their imaging session duration not because of a commonly reported technical limitation, such as photobleaching (which I honestly assumed), but rather the general wellbeing of the animals, who exhibited signs of distress after longer imaging periods. I find this to be a critical issue and perhaps the best argument against performing longer imaging experiments (which would have increased the number of trials, thus potentially boosting the performance of their model). To say that they put animal welfare above all other scientific and technical considerations speaks to a strong ethical adherence to animal welfare policy, and I believe this should be somehow incorporated into the methods.

We have now included this at the top of page 26:

“Mice fully recovered from the brief isoflurane anesthesia, showing a clear blinking reflex, whisking and sniffing behaviors and normal body posture and movements, immediately after head fixation. In our experimental conditions, mice were imaged in sessions of up to 25 min since beyond this time we started observing some signs of distress or discomfort. Thus, we avoided longer recording times at the expense of collecting larger trial numbers, in strong adherence of animal welfare and ethics policy. A pilot group of mice were habituated to the head fixed condition in daily 20 min sessions for 3 days, however we did not observe a marked contrast in the behavior of habituated versus unhabituated mice beyond our relatively short 25 min imaging sessions. In consequence imaging sessions never surpassed a maximum of 25 min, after which the mouse was returned to its home cage.”

(2) Author response image 2: I sincerely thank the authors for providing us reviewers with this figure, which compares the performance of the naïve Bayesian classifier their ultimately use in the study with other commonly implemented models. Also here I falsely assumed that other models, which take correlated activity into account, did not generally perform better than their ultimate model of choice. Although dwelling on it would be distractive (and outside the primary scope of the study), I would encourage the authors to include it as a figure supplement (and simply mention these controls en passant when they justify their choice of the naïve Bayesian classifier).

This figure was now included in the revised manuscript as Figure 2―figure supplement 1.

Page 10 now reads:

“We performed cross-validated, multi-class classification of the single-trial population responses (decoding, Fig. 2A) using a naive Bayes classifier to evaluate the prediction errors as the absolute difference between the stimulus azimuth and the predicted azimuth (Fig. 2A). We chose this classification algorithm over others due to its generally good performance with limited available data. We visualized the cross-validated prediction error distribution in cumulative plots where the observed prediction errors were compared to the distribution of errors for random azimuth sampling (Fig. 2B). When decoding all simultaneously recorded units, the observed classifier output was not significantly better (shifted towards smaller prediction errors) than the chance level distribution (Fig. 2B). The classifier also failed to decode complete DCIC population responses recorded with neuropixels probes (Fig. 3A). Other classifiers performed similarly (Figure 2―figure supplement 1A).”

The bottom paragraph in page 19 now reads:

“To characterize how the observed positive noise correlations could affect the representation of stimulus azimuth by DCIC top ranked unit population responses, we compared the decoding performance obtained by classifying the single-trial response patterns from top ranked units in the modeled decorrelated datasets versus the acquired data (with noise correlations). With the intention to characterize this with a conservative approach that would be less likely to find a contribution of noise correlations as it assumes response independence, we relied on the naive Bayes classifier for decoding throughout the study. Using this classifier, we observed that the modeled decorrelated datasets produced stimulus azimuth prediction error distributions that were significantly shifted towards higher decoding errors (Fig. 6B, C) and, in our imaging datasets, were not significantly different from chance level (Fig. 6B). Altogether, these results suggest that the detected noise correlations in our simultaneously acquired datasets can help reduce the error of the IC population code for sound azimuth. We observed a similar, but not significant tendency with another classifier that does not assume response independence (KNN classifier), though overall producing larger decoding errors than the Bayes classifier (Figure 2―figure supplement 1B).”

**Reviewer #3 (Recommendations for the authors):**
I am generally happy with the response to the reviews.I find the Author response image 3 quite interesting. The neuropixel data looks somewhat like I expected (especially for mouse #3 and maybe mouse #4). I find the distribution of weights across units in the imaging dataset compared to in the pixel dataset intriguing (though it probably is just the dimensionality of the data being so much higher).I'm not too familiar with facial movements but is it the case that the DCIC would be more modulated by ipsilateral movement compared to contralateral movements? Are face movements in mice conjugate or do both sides of the face move more or less independently? If not it may be interesting in future work to record bilaterally and see if that provides more information about DCIC responses.

We sincerely thank the editors and reviewers for their careful appraisal, commendation of our effort and helpful constructive feedback which greatly improved the presentation of our study. Below in green font is a point by point reply to the comments provided by the reviewers.

**Public Reviews:**

**Reviewer #1 (Public Review):**
Summary: In this study, the authors address whether the dorsal nucleus of the inferior colliculus (DCIC) in mice encodes sound source location within the front horizontal plane (i.e., azimuth). They do this using volumetric two-photon Ca2+ imaging and high-density silicon probes (Neuropixels) to collect single-unit data. Such recordings are beneficial because they allow large populations of simultaneous neural data to be collected. Their main results and the claims about those results are the following:(1) DCIC single-unit responses have high trial-to-trial variability (i.e., neural noise);(2) approximately 32% to 40% of DCIC single units have responses that are sensitive to sound source azimuth;(3) single-trial population responses (i.e., the joint response across all sampled single units in an animal) encode sound source azimuth "effectively" (as stated in title) in that localization decoding error matches average mouse discrimination thresholds;(4) DCIC can encode sound source azimuth in a similar format to that in the central nucleus of the inferior colliculus (as stated in Abstract);(5) evidence of noise correlation between pairs of neurons exists;and (6) noise correlations between responses of neurons help reduce population decoding error.While simultaneous recordings are not necessary to demonstrate results #1, #2, and #4, they are necessary to demonstrate results #3, #5, and #6.Strengths:- Important research question to all researchers interested in sensory coding in the nervous system.- State-of-the-art data collection: volumetric two-photon Ca2+ imaging and extracellular recording using high-density probes. Large neuronal data sets.- Confirmation of imaging results (lower temporal resolution) with more traditional microelectrode results (higher temporal resolution).- Clear and appropriate explanation of surgical and electrophysiological methods. I cannot comment on the appropriateness of the imaging methods.Strength of evidence for claims of the study:(1) DCIC single-unit responses have high trial-to-trial variability - The authors' data clearly shows this.(2) Approximately 32% to 40% of DCIC single units have responses that are sensitive to sound source azimuth - The sensitivity of each neuron's response to sound source azimuth was tested with a Kruskal-Wallis test, which is appropriate since response distributions were not normal. Using this statistical test, only 8% of neurons (median for imaging data) were found to be sensitive to azimuth, and the authors noted this was not significantly different than the false positive rate. The Kruskal-Wallis test was not performed on electrophysiological data. The authors suggested that low numbers of azimuth-sensitive units resulting from the statistical analysis may be due to the combination of high neural noise and relatively low number of trials, which would reduce statistical power of the test. This may be true, but if single-unit responses were moderately or strongly sensitive to azimuth, one would expect them to pass the test even with relatively low statistical power. At best, if their statistical test missed some azimuthsensitive units, they were likely only weakly sensitive to azimuth. The authors went on to perform a second test of azimuth sensitivity-a chi-squared test-and found 32% (imaging) and 40% (e-phys) of single units to have statistically significant sensitivity. This feels a bit like fishing for a lower p-value. The Kruskal-Wallis test should have been left as the only analysis. Moreover, the use of a chi-squared test is questionable because it is meant to be used between two categorical variables, and neural response had to be binned before applying the test.

The determination of what is a physiologically relevant “moderate or strong azimuth sensitivity” is not trivial, particularly when comparing tuning across different relays of the auditory pathway like the CNIC, auditory cortex, or in our case DCIC, where physiologically relevant azimuth sensitivities might be different. This is likely the reason why azimuth sensitivity has been defined in diverse ways across the bibliography (see Groh, Kelly & Underhill, 2003 for an early discussion of this issue). These diverse approaches include reaching a certain percentage of maximal response modulation, like used by Day et al. (2012, 2015, 2016) in CNIC, and ANOVA tests, like used by Panniello et al. (2018) and Groh, Kelly & Underhill (2003) in auditory cortex and IC respectively. Moreover, the influence of response variability and biases in response distribution estimation due to limited sampling has not been usually accounted for in the determination of azimuth sensitivity.

As Reviewer #1 points out, in our study we used an appropriate ANOVA test (KruskalWallis) as a starting point to study response sensitivity to stimulus azimuth at DCIC. Please note that the alpha = 0.05 used for this test is not based on experimental evidence about physiologically relevant azimuth sensitivity but instead is an arbitrary p-value threshold. Using this test on the electrophysiological data, we found that ~ 21% of the simultaneously recorded single units reached significance (n = 4 mice). Nevertheless these percentages, in our small sample size (n = 4) were not significantly different from our false positive detection rate (p = 0.0625, Mann-Whitney, See Author response image 1). In consequence, for both our imaging (Fig. 3C) and electrophysiological data, we could not ascertain if the percentage of neurons reaching significance in these ANOVA tests were indeed meaningfully sensitive to azimuth or this was due to chance.

**Author response image 1. sa4fig1:** Percentage of the neuropixels recorded DCIC single units across mice that showed significant median response tuning, compared to false positive detection rate (α = 0.05, chance level).

We reasoned that the observed markedly variable responses from DCIC units, which frequently failed to respond in many trials (Fig. 3D, 4A), in combination with the limited number of trial repetitions we could collect, results in under-sampled response distribution estimations. This under-sampling can bias the determination of stochastic dominance across azimuth response samples in Kruskal-Wallis tests. We would like to highlight that we decided not to implement resampling strategies to artificially increase the azimuth response sample sizes with “virtual trials”, in order to avoid “fishing for a smaller p-value”, when our collected samples might not accurately reflect the actual response population variability.

As an alternative to hypothesis testing based on ranking and determining stochastic dominance of one or more azimuth response samples (Kruskal-Wallis test), we evaluated the overall statistical dependency to stimulus azimuth of the collected responses. To do this we implement the Chi-square test by binning neuronal responses into categories. Binning responses into categories can reduce the influence of response variability to some extent, which constitutes an advantage of the Chi-square approach, but we note the important consideration that these response categories are arbitrary.

Altogether, we acknowledge that our Chi-square approach to define azimuth sensitivity is not free of limitations and despite enabling the interrogation of azimuth sensitivity at DCIC, its interpretability might not extend to other brain regions like CNIC or auditory cortex. Nevertheless we hope the aforementioned arguments justify why the Kruskal-Wallis test simply could not “have been left as the only analysis”.

(3) Single-trial population responses encode sound source azimuth "effectively" in that localization decoding error matches average mouse discrimination thresholds - If only one neuron in a population had responses that were sensitive to azimuth, we would expect that decoding azimuth from observation of that one neuron's response would perform better than chance. By observing the responses of more than one neuron (if more than one were sensitive to azimuth), we would expect performance to increase. The authors found that decoding from the whole population response was no better than chance. They argue (reasonably) that this is because of overfitting of the decoder modeltoo few trials used to fit too many parameters-and provide evidence from decoding combined with principal components analysis which suggests that overfitting is occurring. What is troubling is the performance of the decoder when using only a handful of "topranked" neurons (in terms of azimuth sensitivity) (Fig. 4F and G). Decoder performance seems to increase when going from one to two neurons, then decreases when going from two to three neurons, and doesn't get much better for more neurons than for one neuron alone. It seems likely there is more information about azimuth in the population response, but decoder performance is not able to capture it because spike count distributions in the decoder model are not being accurately estimated due to too few stimulus trials (14, on average). In other words, it seems likely that decoder performance is underestimating the ability of the DCIC population to encode sound source azimuth.To get a sense of how effective a neural population is at coding a particular stimulus parameter, it is useful to compare population decoder performance to psychophysical performance. Unfortunately, mouse behavioral localization data do not exist. Therefore, the authors compare decoder error to mouse left-right discrimination thresholds published previously by a different lab. However, this comparison is inappropriate because the decoder and the mice were performing different perceptual tasks. The decoder is classifying sound sources to 1 of 13 locations from left to right, whereas the mice were discriminating between left or right sources centered around zero degrees. The errors in these two tasks represent different things. The two data sets may potentially be more accurately compared by extracting information from the confusion matrices of population decoder performance. For example, when the stimulus was at -30 deg, how often did the decoder classify the stimulus to a lefthand azimuth? Likewise, when the stimulus was +30 deg, how often did the decoder classify the stimulus to a righthand azimuth?

The azimuth discrimination error reported by Lauer et al. (2011) comes from engaged and highly trained mice, which is a very different context to our experimental setting with untrained mice passively listening to stimuli from 13 random azimuths. Therefore we did not perform analyses or interpretations of our results based on the behavioral task from Lauer et al. (2011) and only made the qualitative observation that the errors match for discussion.

We believe it is further important to clarify that Lauer et al. (2011) tested the ability of mice to discriminate between a positively conditioned stimulus (reference speaker at 0º center azimuth associated to a liquid reward) and a negatively conditioned stimulus (coming from one of five comparison speakers positioned at 20º, 30º, 50º, 70 and 90º azimuth, associated to an electrified lickport) in a conditioned avoidance task. In this task, mice are not precisely “discriminating between left or right sources centered around zero degrees”, making further analyses to compare the experimental design of Lauer et al (2011) and ours even more challenging for valid interpretation.

(4) DCIC can encode sound source azimuth in a similar format to that in the central nucleus of the inferior colliculus - It is unclear what exactly the authors mean by this statement in the Abstract. There are major differences in the encoding of azimuth between the two neighboring brain areas: a large majority of neurons in the CNIC are sensitive to azimuth (and strongly so), whereas the present study shows a minority of azimuth-sensitive neurons in the DCIC. Furthermore, CNIC neurons fire reliably to sound stimuli (low neural noise), whereas the present study shows that DCIC neurons fire more erratically (high neural noise).

Since sound source azimuth is reported to be encoded by population activity patterns at CNIC (Day and Delgutte, 2013), we refer to a population activity pattern code as the “similar format” in which this information is encoded at DCIC. Please note that this is a qualitative comparison and we do not claim this is the “same format”, due to the differences the reviewer precisely describes in the encoding of azimuth at CNIC where a much larger majority of neurons show stronger azimuth sensitivity and response reliability with respect to our observations at DCIC. By this qualitative similarity of encoding format we specifically mean the similar occurrence of activity patterns from azimuth sensitive subpopulations of neurons in both CNIC and DCIC, which carry sufficient information about the stimulus azimuth for a sufficiently accurate prediction with regard to the behavioral discrimination ability.

(5) Evidence of noise correlation between pairs of neurons exists - The authors' data and analyses seem appropriate and sufficient to justify this claim.(6) Noise correlations between responses of neurons help reduce population decoding error - The authors show convincing analysis that performance of their decoder increased when simultaneously measured responses were tested (which include noise correlation) than when scrambled-trial responses were tested (eliminating noise correlation). This makes it seem likely that noise correlation in the responses improved decoder performance. The authors mention that the naïve Bayesian classifier was used as their decoder for computational efficiency, presumably because it assumes no noise correlation and, therefore, assumes responses of individual neurons are independent of each other across trials to the same stimulus. The use of decoder that assumes independence seems key here in testing the hypothesis that noise correlation contains information about sound source azimuth. The logic of using this decoder could be more clearly spelled out to the reader. For example, if the null hypothesis is that noise correlations do not carry azimuth information, then a decoder that assumes independence should perform the same whether population responses are simultaneous or scrambled. The authors' analysis showing a difference in performance between these two cases provides evidence against this null hypothesis.

We sincerely thank the reviewer for this careful and detailed consideration of our analysis approach. Following the reviewer’s constructive suggestion, we justified the decoder choice in the results section at the last paragraph of page 18:

“To characterize how the observed positive noise correlations could affect the representation of stimulus azimuth by DCIC top ranked unit population responses, we compared the decoding performance obtained by classifying the single-trial response patterns from top ranked units in the modeled decorrelated datasets versus the acquired data (with noise correlations). With the intention to characterize this with a conservative approach that would be less likely to find a contribution of noise correlations as it assumes response independence, we relied on the naive Bayes classifier for decoding throughout the study.

Using this classifier, we observed that the modeled decorrelated datasets produced stimulus azimuth prediction error distributions that were significantly shifted towards higher decoding errors (Fig. 5B, C) and, in our imaging datasets, were not significantly different from chance level (Fig. 5B). Altogether, these results suggest that the detected noise correlations in our simultaneously acquired datasets can help reduce the error of the IC population code for sound azimuth.”

Minor weakness:- Most studies of neural encoding of sound source azimuth are done in a noise-free environment, but the experimental setup in the present study had substantial background noise. This complicates comparison of the azimuth tuning results in this study to those of other studies. One is left wondering if azimuth sensitivity would have been greater in the absence of background noise, particularly for the imaging data where the signal was only about 12 dB above the noise. The description of the noise level and signal + noise level in the Methods should be made clearer. Mice hear from about 2.5 - 80 kHz, so it is important to know the noise level within this band as well as specifically within the band overlapping with the signal.

We agree with the reviewer that this information is useful. In our study, the background R.M.S. SPL during imaging across the mouse hearing range (2.5-80kHz) was 44.53 dB and for neuropixels recordings 34.68 dB. We have added this information to the methods section of the revised manuscript.

**Reviewer #2 (Public Review):**
In the present study, Boffi et al. investigate the manner in which the dorsal cortex of the of the inferior colliculus (DCIC), an auditory midbrain area, encodes sound location azimuth in awake, passively listening mice. By employing volumetric calcium imaging (scanned temporal focusing or s-TeFo), complemented with high-density electrode electrophysiological recordings (neuropixels probes), they show that sound-evoked responses are exquisitely noisy, with only a small portion of neurons (units) exhibiting spatial sensitivity. Nevertheless, a naïve Bayesian classifier was able to predict the presented azimuth based on the responses from small populations of these spatially sensitive units. A portion of the spatial information was provided by correlated trial-to-trial response variability between individual units (noise correlations). The study presents a novel characterization of spatial auditory coding in a non-canonical structure, representing a noteworthy contribution specifically to the auditory field and generally to systems neuroscience, due to its implementation of state-of-the-art techniques in an experimentally challenging brain region. However, nuances in the calcium imaging dataset and the naïve Bayesian classifier warrant caution when interpreting some of the results.Strengths:The primary strength of the study lies in its methodological achievements, which allowed the authors to collect a comprehensive and novel dataset. While the DCIC is a dorsal structure, it extends up to a millimetre in depth, making it optically challenging to access in its entirety. It is also more highly myelinated and vascularised compared to e.g., the cerebral cortex, compounding the problem. The authors successfully overcame these challenges and present an impressive volumetric calcium imaging dataset. Furthermore, they corroborated this dataset with electrophysiological recordings, which produced overlapping results. This methodological combination ameliorates the natural concerns that arise from inferring neuronal activity from calcium signals alone, which are in essence an indirect measurement thereof.Another strength of the study is its interdisciplinary relevance. For the auditory field, it represents a significant contribution to the question of how auditory space is represented in the mammalian brain. "Space" per se is not mapped onto the basilar membrane of the cochlea and must be computed entirely within the brain. For azimuth, this requires the comparison between miniscule differences between the timing and intensity of sounds arriving at each ear. It is now generally thought that azimuth is initially encoded in two, opposing hemispheric channels, but the extent to which this initial arrangement is maintained throughout the auditory system remains an open question. The authors observe only a slight contralateral bias in their data, suggesting that sound source azimuth in the DCIC is encoded in a more nuanced manner compared to earlier processing stages of the auditory hindbrain. This is interesting, because it is also known to be an auditory structure to receive more descending inputs from the cortex.Systems neuroscience continues to strive for the perfection of imaging novel, less accessible brain regions. Volumetric calcium imaging is a promising emerging technique, allowing the simultaneous measurement of large populations of neurons in three dimensions. But this necessitates corroboration with other methods, such as electrophysiological recordings, which the authors achieve. The dataset moreover highlights the distinctive characteristics of neuronal auditory representations in the brain. Its signals can be exceptionally sparse and noisy, which provide an additional layer of complexity in the processing and analysis of such datasets. This will be undoubtedly useful for future studies of other less accessible structures with sparse responsiveness.Weaknesses:Although the primary finding that small populations of neurons carry enough spatial information for a naïve Bayesian classifier to reasonably decode the presented stimulus is not called into question, certain idiosyncrasies, in particular the calcium imaging dataset and model, complicate specific interpretations of the model output, and the readership is urged to interpret these aspects of the study's conclusions with caution.I remain in favour of volumetric calcium imaging as a suitable technique for the study, but the presently constrained spatial resolution is insufficient to unequivocally identify regions of interest as cell bodies (and are instead referred to as "units" akin to those of electrophysiological recordings). It remains possible that the imaging set is inadvertently influenced by non-somatic structures (including neuropil), which could report neuronal activity differently than cell bodies. Due to the lack of a comprehensive ground-truth comparison in this regard (which to my knowledge is impossible to achieve with current technology), it is difficult to imagine how many informative such units might have been missed because their signals were influenced by spurious, non-somatic signals, which could have subsequently misled the models. The authors reference the original Nature Methods article (Prevedel et al., 2016) throughout the manuscript, presumably in order to avoid having to repeat previously published experimental metrics. But the DCIC is neither the cortex nor hippocampus (for which the method was originally developed) and may not have the same light scattering properties (not to mention neuronal noise levels). Although the corroborative electrophysiology data largely eleviates these concerns for this particular study, the readership should be cognisant of such caveats, in particular those who are interested in implementing the technique for their own research.A related technical limitation of the calcium imaging dataset is the relatively low number of trials (14) given the inherently high level of noise (both neuronal and imaging). Volumetric calcium imaging, while offering a uniquely expansive field of view, requires relatively high average excitation laser power (in this case nearly 200 mW), a level of exposure the authors may have wanted to minimise by maintaining a low the number of repetitions, but I yield to them to explain.

We assumed that the levels of heating by excitation light measured at the neocortex in Prevedel et al. (2016), were representative for DCIC also. Nevertheless, we recognize this approximation might not be very accurate, due to the differences in tissue architecture and vascularization from these two brain areas, just to name a few factors. The limiting factor preventing us from collecting more trials in our imaging sessions was that we observed signs of discomfort or slight distress in some mice after ~30 min of imaging in our custom setup, which we established as a humane end point to prevent distress. In consequence imaging sessions were kept to 25 min in duration, limiting the number of trials collected. However we cannot rule out that with more extensive habituation prior to experiments the imaging sessions could be prolonged without these signs of discomfort or if indeed influence from our custom setup like potential heating of the brain by illumination light might be the causing factor of the observed distress. Nevertheless, we note that previous work has shown that ~200mW average power is a safe regime for imaging in the cortex by keeping brain heating minimal (Prevedel et al., 2016), without producing the lasting damages observed by immunohistochemisty against apoptosis markers above 250mW (Podgorski and Ranganathan 2016, https://doi.org/10.1152/jn.00275.2016).

Calcium imaging is also inherently slow, requiring relatively long inter-stimulus intervals (in this case 5 s). This unfortunately renders any model designed to predict a stimulus (in this case sound azimuth) from particularly noisy population neuronal data like these as highly prone to overfitting, to which the authors correctly admit after a model trained on the entire raw dataset failed to perform significantly above chance level. This prompted them to feed the model only with data from neurons with the highest spatial sensitivity. This ultimately produced reasonable performance (and was implemented throughout the rest of the study), but it remains possible that if the model was fed with more repetitions of imaging data, its performance would have been more stable across the number of units used to train it. (All models trained with imaging data eventually failed to converge.) However, I also see these limitations as an opportunity to improve the technology further, which I reiterate will be generally important for volume imaging of other sparse or noisy calcium signals in the brain.Transitioning to the naïve Bayesian classifier itself, I first openly ask the authors to justify their choice of this specific model. There are countless types of classifiers for these data, each with their own pros and cons. Did they actually try other models (such as support vector machines), which ultimately failed? If so, these negative results (even if mentioned en passant) would be extremely valuable to the community, in my view. I ask this specifically because different methods assume correspondingly different statistical properties of the input data, and to my knowledge naïve Bayesian classifiers assume that predictors (neuronal responses) are assumed to be independent within a class (azimuth). As the authors show that noise correlations are informative in predicting azimuth, I wonder why they chose a model that doesn't take advantage of these statistical regularities. It could be because of technical considerations (they mention computing efficiency), but I am left generally uncertain about the specific logic that was used to guide the authors through their analytical journey.

One of the main reasons we chose the naïve Bayesian classifier is indeed because it assumes that the responses of the simultaneously recorded neurons are independent and therefore it does not assume a contribution of noise correlations to the estimation of the posterior probability of each azimuth. This model would represent the null hypothesis that noise correlations do not contribute to the encoding of stimulus azimuth, which would be verified by an equal decoding outcome from correlated or decorrelated datasets. Since we observed that this is not the case, the model supports the alternative hypothesis that noise correlations do indeed influence stimulus azimuth encoding. We wanted to test these hypotheses with the most conservative approach possible that would be least likely to find a contribution of noise correlations. Other relevant reasons that justify our choice of the naive Bayesian classifier are its robustness against the limited numbers of trials we could collect in comparison to other more “data hungry” classifiers like SVM, KNN, or artificial neuronal nets. We did perform preliminary tests with alternative classifiers but the obtained decoding errors were similar when decoding the whole population activity (Figure 2―figure supplement 1A). Dimensionality reduction following the approach described in the manuscript showed a tendency towards smaller decoding errors observed with an alternative classifier like KNN, but these errors were still larger than the ones observed with the naive Bayesian classifier (median error 45º). Nevertheless, we also observe a similar tendency for slightly larger decoding errors in the absence of noise correlations (decorrelated, Figure 2―figure supplement 1B). Sentences detailing the logic of classifier choice are now included in the results section at page 10 and at the last paragraph of page 18 (see responses to Reviewer 1).

That aside, there remain other peculiarities in model performance that warrant further investigation. For example, what spurious features (or lack of informative features) in these additional units prevented the models of imaging data from converging?

Considering the amount of variability observed throughout the neuronal responses both in imaging and neuropixels datasets, it is easy to suspect that the information about stimulus azimuth carried in different amounts by individual DCIC neurons can be mixed up with information about other factors (Stringer et al., 2019). In an attempt to study the origin of these features that could confound stimulus azimuth decoding we explored their relation to face movement (Figure 1―figure supplement 2), finding a correlation to snout movements, in line with previous work by Stringer et al. (2019).

In an orthogonal question, did the most spatially sensitive units share any detectable tuning features? A different model trained with electrophysiology data in contrast did not collapse in the range of top-ranked units plotted. Did this model collapse at some point after adding enough units, and how well did that correlate with the model for the imaging data?

Our electrophysiology datasets were much smaller in size (number of simultaneously recorded neurons) compared to our volumetric calcium imaging datasets, resulting in a much smaller total number of top ranked units detected per dataset. This precluded the determination of a collapse of decoder performance due to overfitting beyond the range plotted in Fig 4G.

How well did the form (and diversity) of the spatial tuning functions as recorded with electrophysiology resemble their calcium imaging counterparts? These fundamental questions could be addressed with more basic, but transparent analyses of the data (e.g., the diversity of spatial tuning functions of their recorded units across the population). Even if the model extracts features that are not obvious to the human eye in traditional visualisations, I would still find this interesting.

The diversity of the azimuth tuning curves recorded with calcium imaging (Fig. 3B) was qualitatively larger than the ones recorded with electrophysiology (Fig. 4B), potentially due to the larger sampling obtained with volumetric imaging. We did not perform a detailed comparison of the form and a more quantitative comparison of the diversity of these functions because the signals compared are quite different, as calcium indicator signal is subject to non linearities due to Ca2+ binding cooperativity and low pass filtering due to binding kinetics. We feared this could lead to misleading interpretations about the similarities or differences between the azimuth tuning functions in imaged and electrophysiology datasets. Our model uses statistical response dependency to stimulus azimuth, which does not rely on features from a descriptive statistic like mean response tuning. In this context, visualizing the trial-to-trial responses as a function of azimuth shows “features that are not obvious to the human eye in traditional visualizations” (Fig. 3D, left inset).

Finally, the readership is encouraged to interpret certain statements by the authors in the current version conservatively. How the brain ultimately extracts spatial neuronal data for perception is anyone's guess, but it is important to remember that this study only shows that a naïve Bayesian classifier could decode this information, and it remains entirely unclear whether the brain does this as well. For example, the model is able to achieve a prediction error that corresponds to the psychophysical threshold in mice performing a discrimination task (~30 {degree sign}). Although this is an interesting coincidental observation, it does not mean that the two metrics are necessarily related. The authors correctly do not explicitly claim this, but the manner in which the prose flows may lead a non-expert into drawing that conclusion.

To avoid misleading the non-expert readers, we have clarified in the manuscript that the observed correspondence between decoding error and psychophysical threshold is explicitly coincidental.

Page 13, end of middle paragraph:

“If we consider the median of the prediction error distribution as an overall measure of decoding performance, the single-trial response patterns from subsamples of at least the 7 top ranked units produced median decoding errors that coincidentally matched the reported azimuth discrimination ability of mice (Fig 4G, minimum audible angle = 31º) (Lauer et al., 2011).”

Page 14, bottom paragraph:

“Decoding analysis (Fig. 4F) of the population response patterns from azimuth dependent top ranked units simultaneously recorded with neuropixels probes showed that the 4 top ranked units are the smallest subsample necessary to produce a significant decoding performance that coincidentally matches the discrimination ability of mice (31° (Lauer et al., 2011)) (Fig. 5F, G).”

We also added to the Discussion sentences clarifying that a relationship between these two variables remains to be determined and it also remains to be determined if the DCIC indeed performs a bayesian decoding computation for sound localization.

Page 20, bottom:

“… Concretely, we show that sound location coding does indeed occur at DCIC on the single trial basis, and that this follows a comparable mechanism to the characterized population code at CNIC (Day and Delgutte, 2013). However, it remains to be determined if indeed the DCIC network is physiologically capable of Bayesian decoding computations. Interestingly, the small number of DCIC top ranked units necessary to effectively decode stimulus azimuth suggests that sound azimuth information is redundantly distributed across DCIC top ranked units, which points out that mechanisms beyond coding efficiency could be relevant for this population code.

While the decoding error observed from our DCIC datasets obtained in passively listening, untrained mice coincidentally matches the discrimination ability of highly trained, motivated mice (Lauer et al., 2011), a relationship between decoding error and psychophysical performance remains to be determined. Interestingly, a primary sensory representations should theoretically be even more precise than the behavioral performance as reported in the visual system (Stringer et al., 2021).”

Moreover, the concept of redundancy (of spatial information carried by units throughout the DCIC) is difficult for me to disentangle. One interpretation of this formulation could be that there are non-overlapping populations of neurons distributed across the DCIC that each could predict azimuth independently of each other, which is unlikely what the authors meant. If the authors meant generally that multiple neurons in the DCIC carry sufficient spatial information, then a single neuron would have been able to predict sound source azimuth, which was not the case. I have the feeling that they actually mean "complimentary", but I leave it to the authors to clarify my confusion, should they wish.

We observed that the response patterns from relatively small fractions of the azimuth sensitive DCIC units (4-7 top ranked units) are sufficient to generate an effective code for sound azimuth, while 32-40% of all simultaneously recorded DCIC units are azimuth sensitive. In light of this observation, we interpreted that the azimuth information carried by the population should be redundantly distributed across the complete subpopulation of azimuth sensitive DCIC units.

In summary, the present study represents a significant body of work that contributes substantially to the field of spatial auditory coding and systems neuroscience. However, limitations of the imaging dataset and model as applied in the study muddles concrete conclusions about how the DCIC precisely encodes sound source azimuth and even more so to sound localisation in a behaving animal. Nevertheless, it presents a novel and unique dataset, which, regardless of secondary interpretation, corroborates the general notion that auditory space is encoded in an extraordinarily complex manner in the mammalian brain.
**Reviewer #3 (Public Review):**
Summary: Boffi and colleagues sought to quantify the single-trial, azimuthal information in the dorsal cortex of the inferior colliculus (DCIC), a relatively understudied subnucleus of the auditory midbrain. They used two complementary recording methods while mice passively listened to sounds at different locations: a large volume but slow sampling calcium-imaging method, and a smaller volume but temporally precise electrophysiology method. They found that neurons in the DCIC were variable in their activity, unreliably responding to sound presentation and responding during inter-sound intervals. Boffi and colleagues used a naïve Bayesian decoder to determine if the DCIC population encoded sound location on a single trial. The decoder failed to classify sound location better than chance when using the raw single-trial population response but performed significantly better than chance when using intermediate principal components of the population response. In line with this, when the most azimuth dependent neurons were used to decode azimuthal position, the decoder performed equivalently to the azimuthal localization abilities of mice. The top azimuthal units were not clustered in the DCIC, possessed a contralateral bias in response, and were correlated in their variability (e.g., positive noise correlations). Interestingly, when these noise correlations were perturbed by inter-trial shuffling decoding performance decreased. Although Boffi and colleagues display that azimuthal information can be extracted from DCIC responses, it remains unclear to what degree this information is used and what role noise correlations play in azimuthal encoding.Strengths: The authors should be commended for collection of this dataset. When done in isolation (which is typical), calcium imaging and linear array recordings have intrinsic weaknesses. However, those weaknesses are alleviated when done in conjunction with one another - especially when the data largely recapitulates the findings of the other recording methodology. In addition to the video of the head during the calcium imaging, this data set is extremely rich and will be of use to those interested in the information available in the DCIC, an understudied but likely important subnucleus in the auditory midbrain.The DCIC neural responses are complex; the units unreliably respond to sound onset, and at the very least respond to some unknown input or internal state (e.g., large inter-sound interval responses). The authors do a decent job in wrangling these complex responses: using interpretable decoders to extract information available from population responses.Weaknesses:The authors observe that neurons with the most azimuthal sensitivity within the DCIC are positively correlated, but they use a Naïve Bayesian decoder which assume independence between units. Although this is a bit strange given their observation that some of the recorded units are correlated, it is unlikely to be a critical flaw. At one point the authors reduce the dimensionality of their data through PCA and use the loadings onto these components in their decoder. PCA incorporates the correlational structure when finding the principal components and constrains these components to be orthogonal and uncorrelated. This should alleviate some of the concern regarding the use of the naïve Bayesian decoder because the projections onto the different components are independent. Nevertheless, the decoding results are a bit strange, likely because there is not much linearly decodable azimuth information in the DCIC responses. Raw population responses failed to provide sufficient information concerning azimuth for the decoder to perform better than chance. Additionally, it only performed better than chance when certain principal components or top ranked units contributed to the decoder but not as more components or units were added. So, although there does appear to be some azimuthal information in the recoded DCIC populations - it is somewhat difficult to extract and likely not an 'effective' encoding of sound localization as their title suggests.

As described in the responses to reviewers 1 and 2, we chose the naïve Bayes classifier as a decoder to determine the influence of noise correlations through the most conservative approach possible, as this classifier would be least likely to find a contribution of correlated noise. Also, we chose this decoder due to its robustness against limited numbers of trials collected, in comparison to “data hungry” non linear classifiers like KNN or artificial neuronal nets. Lastly, we observed that small populations of noisy, unreliable (do not respond in every trial) DCIC neurons can encode stimulus azimuth in passively listening mice matching the discrimination error of trained mice. Therefore, while this encoding is definitely not efficient, it can still be considered effective.

Although this is quite a worthwhile dataset, the authors present relatively little about the characteristics of the units they've recorded. This may be due to the high variance in responses seen in their population. Nevertheless, the authors note that units do not respond on every trial but do not report what percent of trials that fail to evoke a response. Is it that neurons are noisy because they do not respond on every trial or is it also that when they do respond they have variable response distributions? It would be nice to gain some insight into the heterogeneity of the responses.

The limited number of azimuth trial repetitions that we could collect precluded us from making any quantification of the unreliability (failures to respond) and variability in the response distributions from the units we recorded, as we feared they could be misleading. In qualitative terms, “due to the high variance in responses seen” in the recordings and the limited trial sampling, it is hard to make any generalization. In consequence we referred to the observed response variance altogether as neuronal noise. Considering these points, our datasets are publicly available for exploration of the response characteristics.

Additionally, is there any clustering at all in response profiles or is each neuron they recorded in the DCIC unique?

We attempted to qualitatively visualize response clustering using dimensionality reduction, observing different degrees of clustering or lack thereof across the azimuth classes in the datasets collected from different mice. It is likely that the limited number of azimuth trials we could collect and the high response variance contribute to an inconsistent response clustering across datasets.

They also only report the noise correlations for their top ranked units, but it is possible that the noise correlations in the rest of the population are different.

For this study, since our aim was to interrogate the influence of noise correlations on stimulus azimuth encoding by DCIC populations, we focused on the noise correlations from the top ranked unit subpopulation, which likely carry the bulk of the sound location information. Noise correlations can be defined as correlation in the trial to trial response variation of neurons. In this respect, it is hard to ascertain if the rest of the population, that is not in the top rank unit percentage, are really responding and showing response variation to evaluate this correlation, or are simply not responding at all and show unrelated activity altogether. This makes observations about noise correlations from “the rest of the population” potentially hard to interpret.

It would also be worth digging into the noise correlations more - are units positively correlated because they respond together (e.g., if unit x responds on trial 1 so does unit y) or are they also modulated around their mean rates on similar trials (e.g., unit x and y respond and both are responding more than their mean response rate). A large portion of trial with no response can occlude noise correlations. More transparency around the response properties of these populations would be welcome.

Due to the limited number of azimuth trial repetitions collected, to evaluate noise correlations we used the non parametric Kendall tau correlation coefficient which is a measure of pairwise rank correlation or ordinal association in the responses to each azimuth. Positive rank correlation would represent neurons more likely responding together. Evaluating response modulation “around their mean rates on similar trials” would require assumptions about the response distributions, which we avoided due to the potential biases associated with limited sample sizes.

It is largely unclear what the DCIC is encoding. Although the authors are interested in azimuth, sound location seems to be only a small part of DCIC responses. The authors report responses during inter-sound interval and unreliable sound-evoked responses. Although they have video of the head during recording, we only see a correlation to snout and ear movements (which are peculiar since in the example shown it seems the head movements predict the sound presentation). Additional correlates could be eye movements or pupil size. Eye movement are of particular interest due to their known interaction with IC responses - especially if the DCIC encodes sound location in relation to eye position instead of head position (though much of eye-position-IC work was done in primates and not rodent). Alternatively, much of the population may only encode sound location if an animal is engaged in a localization task. Ideally, the authors could perform more substantive analyses to determine if this population is truly noisy or if the DCIC is integrating un-analyzed signals.

We unsuccessfully attempted eye tracking and pupillometry in our videos. We suspect that the reason behind this is a generally overly dilated pupil due to the low visible light illumination conditions we used which were necessary to protect the PMT of our custom scope.

It is likely that DCIC population activity is integrating un-analyzed signals, like the signal associated with spontaneous behaviors including face movements (Stringer et al., 2019), which we observed at the level of spontaneous snout movements. However investigating if and how these signals are integrated to stimulus azimuth coding requires extensive behavioral testing and experimentation which is out of the scope of this study. For the purpose of our study, we referred to trial-to-trial response variation as neuronal noise. We note that this definition of neuronal noise can, and likely does, include an influence from un-analyzed signals like the ones from spontaneous behaviors.

Although this critique is ubiquitous among decoding papers in the absence of behavioral or causal perturbations, it is unclear what - if any - role the decoded information may play in neuronal computations. The interpretation of the decoder means that there is some extractable information concerning sound azimuth - but not if it is functional. This information may just be epiphenomenal, leaking in from inputs, and not used in computation or relayed to downstream structures. This should be kept in mind when the authors suggest their findings implicate the DCIC functionally in sound localization.

Our study builds upon previous reports by other independent groups relying on “causal and behavioral perturbations” and implicating DCIC in sound location learning induced experience dependent plasticity (Bajo et al., 2019, 2010; Bajo and King, 2012), which altogether argues in favor of DCIC functionality in sound localization.

Nevertheless, we clarified in the discussion of the revised manuscript that a relationship between the observed decoding error and the psychophysical performance, or the ability of the DCIC network to perform Bayesian decoding computations, both remain to be determined (please see responses to Reviewer #2).

It is unclear why positive noise correlations amongst similarly tuned neurons would improve decoding. A toy model exploring how positive noise correlations in conjunction with unreliable units that inconsistently respond may anchor these findings in an interpretable way. It seems plausible that inconsistent responses would benefit from strong noise correlations, simply by units responding together. This would predict that shuffling would impair performance because you would then be sampling from trials in which some units respond, and trials in which some units do not respond - and may predict a bimodal performance distribution in which some trials decode well (when the units respond) and poor performance (when the units do not respond).

In samples with more that 2 dimensions, the relationship between signal and noise correlations is more complex than in two dimensional samples (Montijn et al., 2016) which makes constructing interpretable and simple toy models of this challenging. Montijn et al. (2016) provide a detailed characterization and model describing how the accuracy of a multidimensional population code can improve when including “positive noise correlations amongst similarly tuned neurons”. Unfortunately we could not successfully test their model based on Mahalanobis distances as we could not verify that the recorded DCIC population responses followed a multivariate gaussian distribution, due to the limited azimuth trial repetitions we could sample.

Significance: Boffi and colleagues set out to parse the azimuthal information available in the DCIC on a single trial. They largely accomplish this goal and are able to extract this information when allowing the units that contain more information about sound location to contribute to their decoding (e.g., through PCA or decoding on top unit activity specifically). The dataset will be of value to those interested in the DCIC and also to anyone interested in the role of noise correlations in population coding. Although this work is first step into parsing the information available in the DCIC, it remains difficult to interpret if/how this azimuthal information is used in localization behaviors of engaged mice.